# Intercomparison of Multiple UV-LIF Spectrometers Using the Aerosol Challenge Simulator

**Elizabeth Forde** [1,*] , **Martin Gallagher** [1] , **Maurice Walker** [2], **Virginia Foot** [2], **Alexis Attwood** [3], **Gary Granger** [3], **Roland Sarda-Estève** [4], **Warren Stanley** [5], **Paul Kaye** [5] **and David Topping** [1]

1   Centre for Atmospheric Science, The University of Manchester, Manchester M13 9PL, UK
2   Defence Science and Technology Laboratory, Porton Down, Salisbury SP4 0JQ, UK
3   Droplet Measurement Technologies, Longmont, CO 80503, USA
4   Laboratoire des Sciences du Climat et de l'Environnement, 91191 Gif-sur-Yvette, France
5   Particle Instruments Research Group, University of Hertfordshire, Hatfield, Hertfordshire AL10 9AB, UK
*   Correspondence: elizabeth.forde@manchester.ac.uk

**Abstract:** Measurements of primary biological aerosol particles (PBAPs) have been conducted worldwide using ultraviolet light-induced fluorescence (UV-LIF) spectrometers. However, how these instruments detect and respond to known biological and non-biological particles, and how they compare, remains uncertain due to limited laboratory intercomparisons. Using the Defence Science and Technology Laboratory, Aerosol Challenge Simulator (ACS), controlled concentrations of biological and non-biological aerosol particles, singly or as mixtures, were produced for testing and intercomparison of multiple versions of the Wideband Integrated Bioaerosol Spectrometer (WIBS) and Multiparameter Bioaerosol Spectrometer (MBS). Although the results suggest some challenges in discriminating biological particle types across different versions of the same UV-LIF instrument, a difference in fluorescence intensity between the non-biological and biological samples could be identified for most instruments. While lower concentrations of fluorescent particles were detected by the MBS, the MBS demonstrates the potential to discriminate between pollen and other biological particles. This study presents the first published technical summary and use of the ACS for instrument intercomparisons. Within this work a clear overview of the data pre-processing is also presented, and documentation of instrument version/model numbers is suggested to assess potential instrument variations between different versions of the same instrument. Further laboratory studies sampling different particle types are suggested before use in quantifying impact on ambient classification.

**Keywords:** PBAP; biological aerosol; bioaerosol; UV-LIF; WIBS; WIBS New Electronic Option (WIBS-NEO); MBS

## 1. Introduction

Primary biological aerosol particles (PBAPs) vary in size and abundance, and comprise viruses (0.01–0.3 μm), bacteria and associated agglomerates (0.1–10 μm), fungal spores (1–30 μm), pollen (5–100 μm) and fragments such as plant and animal debris [1,2]. The dispersal of biological particles in the atmosphere has been found to impact human, animal, and agricultural health [3–5] whilst also affecting the climate and hydrological cycle, by acting as ice nuclei or cloud condensation nuclei [6–8]. Measurements have been conducted worldwide using ultraviolet light-induced fluorescence (UV-LIF) spectrometers, however, laboratory studies quantifying the response of these instruments to known biological and non-biological particle types are limited. Without such studies, assessing and quantifying the emission and impact of these particles on the environment, and plant and human health, will remain restricted.

## 1.1. Overview of PBAP Measurement Techniques

Measurements of PBAP have advanced with developments in technology, which has enabled the production of real-time, online measurement techniques as an alternative to traditional offline techniques for PBAP identification and analysis. Offline techniques, such as impaction onto a collection medium, for instance, using a Hirst spore trap for particle sampling and consequent microscope analysis, remains a commonly used method for pollen particle detection, allowing for identification of the sampled particulates [1,9]. Although the use of offline methods is advantageous compared to online methods, these techniques are often laborious, with poor time resolution, and suffer potential subjectiveness when manually counting particles [1].

UV-LIF spectrometers allow for instantaneous data collection, and work on the principle that biological particles fluoresce when illuminated with ultraviolet light, owing to the intrinsic presence of biofluorophores such as the amino acid, Tryptophan, and the co-enzyme, nicotinamide adenine dinucleotide phosphate (NADH). The detection channels in these instruments are therefore designed to measure the fluorescence coinciding with the maximum emission spectrum of each biofluorophore [10]. Tryptophan is excited at ~280 nm and emits fluorescence between 300 and 400 nm, NADH is excited between 270 and 400 nm and emits between 400 and 600 nm [10,11], and other biofluorophores, such as Riboflavin, are excited at ~450 nm and emit at ~520–565 nm [12,13].

Although there are multiple UV-LIF spectrometers in existence, the most commonly used instrument is the Wideband Integrated Bioaerosol Spectrometer (WIBS), which has evolved over time, to produce a cost-effective yet reliable PBAP detector [10,14,15]. The WIBS works by optical detection of particles sized between 0.5 and 20 μm, encompassing a broad range of particle types including bacteria, fungal spores, smaller-sized pollen or pollen fragments, but not smaller particles such as viruses. The WIBS produces data on particle size and shape, and utilises dual excitation wavebands and three channel detection to identify fluorescence characteristics from sampled particles (discussed further in Section 3.1). A development of the WIBS is the Multiparameter Bioaerosol Spectrometer (MBS) that, as with the WIBS, was developed at the University of Hertfordshire. The MBS features single wavelength excitation and fluorescence detection over eight wavelength bands (300–655 nm), while also similarly detecting particles from 0.5 to 20 μm (discussed further in Section 3.2). The principle of using more detection channels to detect fluorescence following single or dual wavelength excitation is currently being developed. The PLAIR Rapid-E, as based upon the principles described by Kiselev et al. (2011, 2013) [16,17], features fluorescence detection over 32-wavelength bins following single channel excitation, in addition to a larger particle detection range of 1–100 μm. Other examples include the Spectral Intensity Bioaerosol Sensor (SIBS), which features dual wavelength excitation (285 nm and 370 nm) and 16-channel fluorescence detection (302–721 nm) [18–20].

## 1.2. UV-LIF Usage

Ambient measurements have been conducted worldwide using UV-LIF spectrometers, and in comparison to the MBS, for which no published ambient data currently exist, multiple versions of the WIBS have been part of worldwide measurement campaigns [21–28]. Laboratory experiments conducted using UV-LIF spectrometers are limited in relation to the quantity of ambient measurements conducted worldwide. Examples of previous laboratory studies include the detection of pollens and fungal spores using a WIBS-4 [29], and sensitivity tests of the WIBS-4 to both biological samples and potential fluorescent, non-biological interferents [30]. LIF instrumentation has additionally been used to detect biological and other organic-carbon aerosol particles using a BSL-3 facility[31,32]. More recently, work has been conducted using a WIBS-4A and ABC classification of over 50 pure cultures of bacteria, fungi and pollen particulates [33], in addition to analysis of sixty-nine particle types comprising biological and common interferent particles, using a new fluorescent threshold for analysis [34].

*1.3. UV-LIF Discrimination*

Although UV-LIF instrumentation offers various advantages compared to the traditional offline methods, the presence of non-biological particles fluorescing at wavelengths used by UV-LIF instruments, such as combustion type particles [35], present a potential interference risk. There are also some uncertainties concerning the unknown effects from atmospheric transportation and the impact on biological particles, in particular the effects on particle fluorescence [2,36]. In addition, the detection of mixtures of biological particles with non-biological particles, such as bacteria or fungal spores with dust [4,37,38] is unclear, and the interpretation of such events by UV-LIF instruments is uncertain.

*1.4. Scope*

During September 2017, an intensive chamber experiment was conducted using the Aerosol Challenge Simulator (ACS) facility at the Defence Science and Technology Laboratory (Dstl) at Porton Down, Wiltshire, United Kingdom. The ACS was used to produce controlled concentrations of both biological and non-biological particulates for detection by five UV-LIF spectrometers. The instruments used in this study consisted of two Wideband Integrated Bioaerosol Spectrometers (Version 4; WIBS-4), the commercial WIBS-New Electronic Option (WIBS-NEO) as produced by Droplet Measurement Technologies, and two Multiparameter Bioaerosol Spectrometers (MBS). For the first time, an intercomparison of multiple UV-LIF instrumentation, including comparisons of multiple versions of the same instrument, is presented. Using controlled concentrations of known biological and non-biological particle types produced by the ACS, an assessment of the UV-LIF instruments ability and consistency in detecting these particles is demonstrated. Whilst this paper presents the first technical summary and use of the ACS, this study aims to additionally investigate the usefulness of these measurements for ambient dataset interpretation and discrimination algorithm development.

## 2. The Aerosol Challenge Simulator (ACS)

The Aerosol Challenge Simulator (ACS) is installed at the Defence Science and Technology Laboratory at Porton Down in Salisbury, Wiltshire, UK (Figure 1). The ACS is designed to produced computer-controlled concentrations, or "challenges", of known particle types, such as bacteria, pollen and fungal material.

Challenge particles are generated and released from either of two aerosol dissemination sections, which consist of two $0.43$ m$^3$ volume boxes with Ultra Low Particulate Air (ULPA) filters on their inlet air stream. Of the two dissemination sections, one aerosol disseminator generally provides a "background" sample whilst the other provides the "challenge" aerosol. In this study, dry powders were released into the background side of the ACS, and liquid solutions nebulised into the challenge side (Figure 1).

The aerosols pass into dilution sections where the aerosol flow is split, with a proportion extracted and ULPA filtered, and reintroduced. Aerosol particles are monitored by GRIMM optical particle counters (Grimm Aerosol Technik Pouch GmbH: Muldestausee, Germany; discussed further in Section 2.1) which are situated at multiple locations on the ACS. The aerosol concentration and challenge levels are computer-controlled to achieve the required dilution. The outputs from the two diluted dissemination sections are then combined within a third mixing box, and passed to the test section.

Within the test section, the UV-LIF instruments were connected to the manifold tube using isokinetic probes and static dissipative tubing. The instruments were set up to sample prior to challenge release and stopped after the challenge had stopped, when concentrations had become low. The total flow through the test section is approximately 2500 L per minute. The aerosol stream is then passed to a double ULPA exhaust filter section.

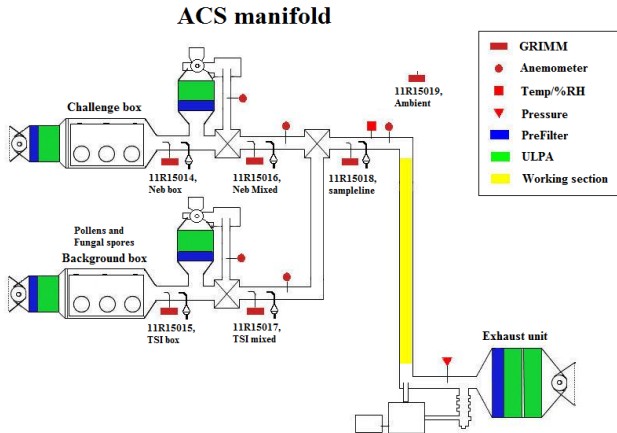

**Figure 1.** Schematic diagram of the Aerosol Challenge Simulator (ACS), UV-LIF (ultraviolet light-induced fluorescence) spectrometers were positioned on the highlighted yellow "working section". (ULPA: Ultra Low Particulate Air).

### 2.1. ACS Concentration Monitoring

Five GRIMM optical particle counters (OPC) are situated at selected points on the ACS, in addition to one set up to detect ambient conditions (Figure 1). The GRIMM OPC (Model number 11-R) detects particles from 0.25 to 32 μm, at a sample flow of 1.2 L/min, and uses a 660 nm laser and detector to measure side-scatter from sampled particles which are then classified into 31 size channels.

### 2.2. Aerosolisation Methods

Dry powders were dispersed using a modified and stripped down TSI Small-scale Powder Disperser (SSPD) (Model No. 3433) set up from within the glove-fronted "background" dissemination box of the ACS. The SSPD sample plate comprised a custom made deep groove plate with three concentric rings, which were filled with the selected particle type, by hand, using a razor blade to move the particulate uniformly into the grooves. The sample plate was initially manually rotated to obtain an adequate aerosolisation rate, but later automated. To disperse the liquid solutions of particles, such as the Polystyrene Latex Spheres (PSLs) and the bacterial particles, a medical nebuliser was set up in the "challenge" dissemination box of the ACS.

### 2.3. Challenge Particle List

The particles sampled included pollens and fungal spore material supplied by Greer, bacteria, non-biological potential "interferent" particles and calibration polystyrene latex spheres (PSLs) of a range of sizes and doping. The particles used in this study, their provenance and generation method are presented according to the different particle type groupings and are provided in Table S1.

The bacterial samples were both "washed" and "unwashed", for example, the Esherishia Coli (E. coli) sample was unwashed, and suspended in a phosphate-buffered saline (PBS) solution, which was also separately sampled during the experiment (Table S1). Washed bacterial spore samples were produced by filtering the bacterial material and re-suspending the particles in distilled water, as opposed to being suspended in the growth medium (unwashed).

It must be noted that both *Alternaria* (Alternaria alternata) and *Cladosporium* (Cladosporium herbarium) were processed fungal spore material samples and therefore not necessarily representative of "real-world" fungal spores. Although the upper detection limit of the WIBS-4 and MBS is ~20 μm (discussed further in Section 3) the ability of these instruments to detect the pollen samples listed in Table S1 was uncertain, owing to the larger size range of the pollen particles sampled. In addition to the biological particles and calibration PSLs, potential interferent particles such as salt and Arizona

Test Dust were also sampled. Most particle types were sampled only once during the experiment, whilst some were sampled over multiple days (discussed further in Section 4.1.1).

*2.4. Scanning Electron Microscopy*

Scanning electron microscope (SEM) images were taken of a subset of the sampled particles listed in Table S1. Samples were observed using a Hitachi SU-3500 SEM operating at 5 kV. Digital images of each sample were acquired and adjusted for brightness and contrast only. Owing to the ~20 μm upper detection limit of most of the UV-LIF instruments, SEM images were taken of the pollen particles to identify any potential particle characteristics which would affect the detection of these particles. Additionally SEM images were taken of the fungal spore material, *Alternaria* and *Cladosporium*. SEM images were taken of samples taken straight from the sample pot, at the front of the ACS after generation, and from the back of the ACS after the sampling ports, to additionally identify any potential fragmentation or settling occurring during particle generation and dispersion.

## 3. UV-LIF Instrumentation

Five UV-LIF spectrometers were connected to the working section of the ACS and consisted of the three Wideband Integrated Bioaerosol Spectrometers and the two Multiparameter Bioaerosol Spectrometers. Although some instruments were replicates of the same model instrument, differences were expected between these, and between the other instruments. A comparison table with an overview of each instrument is presented in Table 1.

*3.1. WIBS-4*

A brief overview of the WIBS instruments is presented here; more detailed descriptions have been previously discussed [39,40]. Two model WIBS-4 instruments were used comprising a WIBS-4M (Manchester) and a WIBS-4D (Dstl). The WIBS operates by using a 635 nm diode laser beam and quadrant detector to produce a particle size value and an asymmetry factor (Af) value to indicate the particle shape. Utilising dual wavelength excitation (280 nm and 370 nm) and three detector channels (310–400 nm and 420–650 nm), potential fluorescence emission from sampled particles is recorded. Particles are drawn into the WIBS-4M at a sample flow rate of 6.8 mL/s, whilst the WIBS-4D has a modified sample flow of 16.6 mL/s by constraining the sheath flow. However, during the experiment, the sample flow of the WIBS-4D was 20.1 mL/s caused by a blockage of the sheath flow, which led to an increase in the sample flow and may have affected flashlamp timing to illuminate particles. Some of these measurements were repeated once the blockage had been cleared and are discussed further in Section 4.1.1.

*3.2. WIBS-NEO*

The WIBS-NEO unit used in this study is a prototype of the commercially available instrument produced by Droplet Measurement Technologies. The WIBS-NEO is a development of the WIBS-4 utilising new operating software and featuring a marginally larger sample size range (0.5 to ~30 μm), whilst using the same measurement method as the original WIBS models. However, the WIBS-NEO differs in terms of the lower flow rate, and how it measures optical particle size, in which the scattered light intensity from the 635 nm laser is measured by a photomultiplier tube, not by the quadrant detector, which is only used to calculate an Af value. It must be noted that the prototype unit used in this study differs from the commercialised version of the instrument with respect to both hardware and software, and care should be taken when comparing between units, particularly for absolute fluorescent intensity values (discussed further in Section 6).

**Table 1.** Summary table of the different UV-LIF instruments (WIBS-4M, WIBS-4D, WIBS-NEO, MBS-M, and MBS-D) (WIBS: Wideband Integrated Bioaerosol Spectrometer; MBS: Multiparameter Bioaerosol Spectrometers) deployed during the ACS (ACS: Aerosol Challenge Simulator) chamber experiment.

| Parameter | WIBS-4M | WIBS-4D | WIBS-NEO | MBS-M | MBS-D |
|---|---|---|---|---|---|
| Size range | 0.5–20 μm | 0.5–20 μm | 0.5–30 μm | 0.5–20 μm | 0.5–20 μm |
| Total Flow Rate | 2.5 L/min | 2.5 L/min | 2.1 L/min | 1.5 L/min | 1.5 L/min |
| Sample Flow Rate | 0.38 L/min | 1.2 L/min (usually 1 L/min) | 0.3 L/min | 0.2 L/min | 0.2 L/min |
| Size/shape | 635 nm laser | 635 nm laser | 635 nm laser | 635 nm (size) 637 nm (shape) | 635 nm (size) 637 nm (shape) |
| Size/shape detection | Quadrant PMT | Quadrant PMT | Quadrant PMT | CMOS linear arrays | CMOS linear arrays |
| Fl. Excitation | 280 nm, 370 nm | 280 nm, 370 nm | 280 nm, 370 nm | 280 nm | 280 nm |
| Fl. Detection | 310–400 nm, 420–650 nm | 310–400 nm, 420–650 nm | 310–400 nm, 420–650 nm | 8 channel (310–630 nm) | 8 channel detection (310–630 nm) |

*3.3. Multiparameter Bioaerosol Spectrometer (MBS)*

Two MBS instruments were deployed during the ACS experiment and comprised a MBS-M (Manchester) and a MBS-D (Dstl), which were built at different times by the University of Hertfordshire. Utilising a 635 nm diode laser beam to illuminate sampled particles, a lens assembly and photodiode trigger detector are then used to determine an optical equivalent diameter, and a second 637 nm laser is used to determine particle morphology using two 512-pixel CMOS detector arrays. Utilising single wavelength excitation (280 nm), the resulting fluorescence is detected by an eight-channel photomultiplier tube from 310 to 650 nm.

## 4. Data Format and Analysis Protocol

*4.1. Data Format*

In comparison to the WIBS-NEO, which produces Hierarchical Data Format 5 (HDF5) files, the WIBS-4 and MBS instruments produce comma-separated value (CSV) datafiles. In addition to data collected on particle size, shape and fluorescence response, ancillary data are collected by each instrument, including the relative power of each Xenon flash for the different excitation channels, as well as particle time-of-flight information to eliminate particle coincidence events. Both the WIBS-4 and MBS instruments record time-stamped single particle data as measured in milliseconds from the file start time, while the WIBS-NEO records in seconds from Unix epoch time (seconds since 1 January 1904). With the exception of the difference in fluorescence detection recording between the MBS and WIBS, the most apparent difference between the two instruments concerns the recording of particle shape data. The MBS collects 1024 shape measurements, following irradiation by the 637 nm laser and detection by the two 512-pixel CMOS detector arrays, compared to the WIBS, which produces four outputs for particle shape following irradiation by the 635 nm laser and consequent detection by the quadrant detector.

4.1.1. Forced Trigger (FT) Data

The MBS and WIBS instruments have a forced trigger (FT) function in which measurements are collected when no particles are being drawn into the instrument. During this time, the pump is turned off and the flashlamps fire on the "empty" chamber. The duration of FT measurements differ between the instruments, but for the WIBS-4M this occurs for ~10 s. Using the FT measurement(s), the instrument baselines can be determined and used as a fluorescent threshold for acquisition data analysis.

*4.2. Data Pre-Processing*

The raw data were processed differently depending upon the instrument used, and a basic schematic overview of the data pre-processing is presented in Figure S1. The data for all instruments were analysed using an averaged FT measurement + 3 standard deviations (SD), owing to the lack of data remaining for the MBS when using a FT + 9SD threshold, following Savage et al. (2017) [34]. A greater number of particles were detected by the three WIBS instruments, in comparison to the two MBS instruments. The difference between the number of particles detected and influence from the removal of the FT + 3SD threshold are presented in Section S3 in Supplementary Materials. Additionally, for comparison, the removal of the FT + 9SD baseline for the MBS is presented in Section S4 in Supplementary Materials.

While most particle types were sampled only once during the experiment, the few particle types which were sampled over multiple days were combined and analysed as a complete sample. Prior to which, the fluorescent profiles, and particle size and shape of the samples, were compared. By doing so, only some difference could be seen in the fluorescence profiles of the larger particles which was

suggested to result from the difference in the median sizes recorded (Section S5 in Supplementary Materials).

Due to a partial sheath flow blockage, the WIBS-4D initially sampled at a higher flow rate of 20.1 mL/s, as opposed to the expected 16.6 mL/s, and the trigger threshold was set lower, so that it occasionally triggered on "noise". The data presented here were compared to additional data collected using the ACS following the removal of the sheath flow blockage (returning the sample flow rate to 16.6 mL/s), with the same trigger threshold. For the subset of particles re-sampled, the dominant fluorescent channel was different for two out of the nine samples (Section S6 in Supplementary Materials).

### 4.2.1. WIBS Pre-Processing

The WIBS-NEO HDF5 datafiles were converted to CSV files, and those which failed to convert, owing to missing headers or incomplete lines of data, were excluded from the analysis. To establish a baseline for fluorescence, any intrinsic FT data were removed from the acquisition data files for the three WIBS instruments. All FT files collected per instrument were used as a baseline for fluorescence, owing to the different amounts of FT data collected during the campaign.

### 4.2.2. MBS Pre-Processing

The MBS operating software is automated to find the average of each "block" of FT data and then subtract the baseline from subsequent acquisition data in the datafiles. To establish a baseline + 3SD, the data from the instrument needed to be reverse-engineered, to result in acquisition data with no removed baseline. The trends in FT data were plotted to identify any variance between the different files for the MBS-M and the MBS-D (Section S7 in Supplementary Materials). As a result, the files taken for each different particle type were compiled, and the FT data from these files isolated and used as a baseline + 3SD for each MBS instrument.

## 5. Results

The following results section is split into four parts (subsections) to represent the main results produced. In Section 5.1, the fluorescent profiles of the sampled particles are presented for each instrument, and as a result of particle fragmentation occurring for some samples the effects on fluorescence are discussed (Section 5.1.1). Particle type differences and instrument variation are explored in Section 5.2 and an initial comparison is conducted between the WIBS instruments and the MBS instruments. Comparisons are then conducted for the WIBS and MBS instruments, respectively, and their response to bacteria (Section 5.2.1), fungal spore material (Section 5.2.2), pollen and pollen fragments (Section 5.2.3) and non-biological samples (Section 5.2.4). The relationship among particle size, shape and fluorescence is then presented in Section 5.3 with the relationship between particle size and shape shown for each instrument and particle type group (Section 5.3.1). The instrument responses to a dust–bacterial mixture are then discussed in Section 5.4.

### 5.1. Particle Fluorescence Profiles

For each particle type sampled, the WIBS provides data on the fluorescence emission in the three detector channels (FL1, FL2 and FL3) following dual wavelength excitation, the particle size (μm) and the particle asymmetry factor (Af). The MBS differs by recording the fluorescence emission over eight detection channels (XE1–XE8) following single wavelength excitation, and also similarly records the particle size (μm) and the particle shape. For each particle type sampled, the median values in each detection channel, and the median particle size and shape values are presented in Tables 2 and 3 for the two MBS instruments, Tables 4 and 5 for the two WIBS-4 instruments and Table 6 for the WIBS-NEO.

**Table 2.** Median fluorescence profiles, and median particle size and shape values for the MBS-M.

| MBS-M | XE1 | XE2 | XE3 | XE4 | XE5 | XE6 | XE7 | XE8 | SIZE (μm) | SHAPE |
|---|---|---|---|---|---|---|---|---|---|---|
| PSLs | | | | | | | | | | |
| 3 μm | 507.8 ± 0.0 | 274.9 ± 612.6 | 523.1 ± 576.5 | 77.4 ± 719.1 | 387.0 ± 507.7 | 207.1 ± 290.1 | 266.9 ± 0.0 | 411.6 ± 0.0 | 3.2 ± 1.2 | 12.2 ± 6.6 |
| 2 μm blue | 279.9 ± 71.9 | 287.7 ± 67.2 | 347.9 ± 73.6 | 1538.3 ± 435.8 | 1893.1 ± 298.6 | 1921.3 ± 273.5 | 1448.7 ± 218.0 | 487.9 ± 90.7 | 2.0 ± 0.4 | 12.2 ± 6.6 |
| 1 μm blue | 12.5 ± 15.2 | 18.8 ± 50.6 | 20.9 ± 98.7 | 193.0 ± 100.2 | 1742.8 ± 275.9 | 708.1 ± 221.6 | 95.0 ± 59.3 | 15.0 ± 19.0 | 1.3 ± 0.2 | 12.7 ± 4.6 |
| 3 μm green | 361.1 ± 46.4 | 531.0 ± 86.2 | 629.1 ± 130.9 | 477.1 ± 51.1 | 1895.6 ± 295.7 | 1923.1 ± 213.7 | 1952.3 ± 217.8 | 1806.5 ± 342.2 | 3.2 ± 0.7 | 11.4 ± 7.9 |
| 2 μm green | 113.5 ± 31.8 | 160.6 ± 52.3 | 182.0 ± 74.4 | 119.4 ± 49.9 | 1819.3 ± 120.0 | 1919.1 ± 0.3 | 1955.4 ± 151.7 | 795.8 ± 111.9 | 1.8 ± 0.3 | 10.0 ± 4.0 |
| 1 μm green | 9.9 ± 16.3 | 17.1 ± 23.3 | 18.3 ± 24.1 | 17.4 ± 26.0 | 248.7 ± 130.5 | 819.7 ± 220.2 | 270.1 ± 139.5 | 33.2 ± 58.8 | 1.2 ± 0.3 | 12.5 ± 5.8 |
| 23 μm | | 268.5 ± 0.0 | 903.6 ± 1151.9 | 895.4 ± 1107.3 | 620.0 ± 0.0 | 117.0 ± 157.8 | 13.0 ± 0.0 | 31.1 ± 0.0 | 1.0 ± 0.2 | 15.0 ± 21.2 |
| 11 μm | 52.2 ± 944.6 | 148.4 ± 671.2 | 160.0 ± 577.1 | 154.9 ± 562.3 | 34.8 ± 407.7 | 58.0 ± 305.7 | 26.6 ± 378.9 | 18.6 ± 847.2 | 1.2 ± 1.2 | 17.5 ± 13.0 |
| Pollen | | | | | | | | | | |
| Timothy grass (Pheleum pratense) | 226.1 ± 573.1 | 789.7 ± 779.8 | 916.6 ± 721.1 | 741.5 ± 775.7 | 399.0 ± 840.9 | 261.6 ± 895.6 | 1937.3 ± 953.4 | 344.0 ± 904.2 | 1.1 ± 1.2 | 21.4 ± 13.0 |
| Nettle (Urtica dioica) | 1262.4 ± 441.4 | 1858.5 ± 751.7 | 1809.8 ± 755.9 | 1818.6 ± 740.3 | 1893.1 ± 816.8 | 1917.6 ± 833.8 | 1944.8 ± 827.3 | 1774.7 ± 674.8 | 0.9 ± 1.3 | 30.0 ± 12.5 |
| Sheep Sorrel (Rumex acetosella) | 1108.5 ± 533.3 | 1668.5 ± 821.5 | 1796.7 ± 590.0 | 1822.3 ± 588.8 | 1887.5 ± 765.5 | 1921.6 ± 687.6 | 1942.1 ± 577.2 | 1769.7 ± 521.1 | 0.9 ± 0.7 | 34.6 ± 13.6 |
| Ryegrass (Lolium perenne) | 1225.1 ± 568.0 | 1637.5 ± 627.5 | 1808.4 ± 442.7 | 1817.3 ± 424.0 | 1904.2 ± 732.2 | 1916.6 ± 542.9 | 1960.6 ± 842.6 | 1883.5 ± 571.0 | 0.9 ± 1.2 | 32.4 ± 11.8 |
| White Poplar (Populus alba) | 605.7 ± 835.5 | 124.8 ± 837.3 | 276.2 ± 714.6 | 347.8 ± 739.3 | 216.8 ± 733.9 | 149.5 ± 729.3 | 92.7 ± 828.3 | 70.4 ± 934.2 | 2.9 ± 1.9 | 35.4 ± 21.5 |
| European white birch (Betula pendula) | 100.8 ± 627.9 | 183.9 ± 858.1 | 1049.1 ± 879.0 | 1819.6 ± 868.0 | 1313.0 ± 887.2 | 129.9 ± 862.4 | 256.1 ± 877.2 | 574.3 ± 853.4 | 1.4 ± 1.5 | 32.7 ± 13.6 |
| Olive (Olea europaea) | 51.4 ± 496.0 | 186.3 ± 679.5 | 337.9 ± 677.1 | 202.6 ± 712.9 | 1272.2 ± 829.8 | 1212.7 ± 763.9 | 695.7 ± 607.4 | 266.5 ± 638.2 | 1.3 ± 0.7 | 14.5 ± 11.0 |
| English Oak (Quercus robur) | 812.9 ± 1113.0 | 140.1 ± 1023.7 | 198.9 ± 752.0 | 67.9 ± 490.2 | 33.3 ± 660.1 | 34.8 ± 729.3 | 9.9 ± 726.9 | 465.1 ± 932.3 | 2.6 ± 0.9 | 13.4 ± 10.2 |
| Fungal spores | | | | | | | | | | |
| Cladosporium herbarium | 10.6 ± 80.5 | 113.1 ± 155.8 | 228.0 ± 311.0 | 104.6 ± 153.8 | 44.8 ± 110.1 | 29.6 ± 99.1 | 16.3 ± 80.0 | 12.6 ± 103.6 | 2.9 ± 1.5 | 18.9 ± 10.3 |
| Alternaria alternata | 8.5 ± 178.9 | 29.5 ± 200.7 | 65.1 ± 279.7 | 42.9 ± 326.2 | 27.3 ± 335.6 | 19.4 ± 344.0 | 13.6 ± 293.7 | 10.8 ± 297.2 | 1.8 ± 1.3 | 17.6 ± 10.7 |
| Bacteria | | | | | | | | | | |
| Esherishia coli (E. coli) | 12.4 ± 25.1 | 69.4 ± 148.5 | 112.0 ± 312.6 | 83.0 ± 269.8 | 77.8 ± 216.8 | 63.7 ± 143.7 | 26.4 ± 66.3 | 18.3 ± 43.2 | 1.4 ± 0.6 | 12.2 ± 4.8 |
| Bacillus atrophaeus (BG) (washed) | 9.3 ± 11.3 | 16.9 ± 63.2 | 31.4 ± 101.2 | 15.7 ± 73.2 | 14.3 ± 28.6 | 8.2 ± 24.7 | 8.0 ± 14.4 | 10.2 ± 18.2 | 1.2 ± 0.6 | 11.7 ± 5.5 |
| Bacillus atrophaeus (BG) (unwashed) | 8.8 ± 11.6 | 29.0 ± 47.6 | 76.2 ± 132.8 | 84.9 ± 209.6 | 87.7 ± 197.4 | 53.6 ± 124.9 | 29.9 ± 56.4 | 18.0 ± 25.1 | 1.4 ± 0.6 | 11.8 ± 4.2 |
| Bacillus atrophaeus (BT) (washed) | 5.5 ± 10.0 | 20.3 ± 27.2 | 34.8 ± 49.9 | 15.2 ± 26.4 | 12.6 ± 14.6 | 11.7 ± 12.9 | 6.0 ± 8.3 | 6.2 ± 9.6 | 1.5 ± 0.6 | 12.2 ± 5.0 |
| Bacillus atrophaeus (BT) (unwashed) | 5.6 ± 13.1 | 16.5 ± 23.2 | 34.2 ± 75.9 | 75.9 ± 191.2 | 63.8 ± 231.7 | 53.7 ± 181.9 | 41.3 ± 70.9 | 16.2 ± 30.0 | 1.3 ± 0.6 | 12.4 ± 5.0 |
| Others | | | | | | | | | | |
| Arizona test dust | 4.2 ± 6.0 | 8.5 ± 31.0 | 12.7 ± 141.7 | 11.5 ± 160.3 | 10.9 ± 75.3 | 9.1 ± 26.1 | 8.8 ± 29.0 | 5.8 ± 9.5 | 2.6 ± 1.8 | 18.5 ± 12.2 |
| PBS | 6.0 ± 9.1 | 8.1 ± 22.6 | 15.6 ± 123.5 | 14.9 ± 70.5 | 9.5 ± 8.1 | 8.0 ± 7.9 | 4.4 ± 9.4 | 8.5 ± 8.5 | 1.4 ± 0.8 | 11.9 ± 5.7 |
| Salt (NaCl) | 4.8 ± 8.5 | 13.5 ± 0.7 | 12.2 ± 14.5 | 10.5 ± 8.0 | 9.8 ± 11.2 | 7.8 ± 11.6 | 8.8 ± 10.4 | 7.8 ± 17.9 | 1.9 ± 1.4 | 14.7 ± 9.4 |
| Mixture | 6.0 ± 112.8 | 18.0 ± 63.1 | 44.0 ± 121.3 | 47.2 ± 135.1 | 61.1 ± 162.9 | 46.4 ± 372.5 | 33.7 ± 163.2 | 19.6 ± 133.0 | 1.3 ± 1.0 | 12.2 ± 7.8 |

**Table 3.** Median fluorescence profiles, and median particle size and shape values for the MBS-D.

| MBS-D | XE1 | XE2 | XE3 | XE4 | XE5 | XE6 | XE7 | XE8 | SIZE (µm) | SHAPE |
|---|---|---|---|---|---|---|---|---|---|---|
| **PSLs** | | | | | | | | | | |
| 3 µm | 35.7 ± 41.6 | 364.6 ± 198.1 | 1220.9 ± 582.3 | 105.4 ± 124.2 | 63.4 ± 86.7 | 34.4 ± 54.9 | 23.5 ± 24.3 | 16.9 ± 26.0 | 2.8 ± 1.0 | 15.4 ± 21.6 |
| 2 µm blue | 264.4 ± 138.3 | 468.7 ± 217.2 | 1536.7 ± 350.6 | 1555.7 ± 310.6 | 1626.1 ± 198.7 | 1660.7 ± 148.7 | 1860.4 ± 258.0 | 651.7 ± 214.8 | 2.0 ± 0.4 | 8.0 ± 5.4 |
| 1 µm blue | 24.2 ± 28.7 | 33.5 ± 59.0 | 85.0 ± 116.0 | 1311.3 ± 385.9 | 1598.5 ± 145.8 | 1586.9 ± 359.7 | 145.7 ± 120.9 | 20.9 ± 39.6 | 1.0 ± 0.2 | 12.6 ± 5.9 |
| 3 µm green | 365.9 ± 109.9 | 814.5 ± 236.9 | 1495.5 ± 311.5 | 1587.5 ± 157.8 | 1624.3 ± 0.0 | 1696.4 ± 214.8 | 1868.6 ± 151.5 | 1844.8 ± 107.5 | 2.9 ± 0.5 | 13.1 ± 110.6 |
| 2 µm green | 115.1 ± 71.7 | 189.5 ± 107.7 | 438.0 ± 215.0 | 1430.1 ± 442.6 | 1604.4 ± 25.6 | 1684.6 ± 82.5 | 1860.1 ± 63.0 | 1466.8 ± 322.4 | 1.7 ± 0.2 | 8.1 ± 6.2 |
| 1 µm green | 19.2 ± 24.6 | 36.6 ± 40.4 | 47.3 ± 58.2 | 75.8 ± 91.8 | 949.3 ± 348.5 | 1690.1 ± 258.6 | 574.9 ± 250.4 | 50.3 ± 71.2 | 1.0 ± 0.2 | 13.4 ± 11.1 |
| 23 µm | 81.1 ± 175.8 | 446.3 ± 430.6 | 1451.5 ± 494.4 | 304.5 ± 350.6 | 56.3 ± 74.0 | 59.0 ± 65.8 | 14.8 ± 9.9 | 2.0 ± 0.1 | 3.0 ± 1.4 | 24.1 ± 95.4 |
| 11 µm | 79.0 ± 328.5 | 290.4 ± 468.7 | 689.4 ± 644.7 | 208.0 ± 532.4 | 147.8 ± 522.5 | 139.4 ± 429.7 | 32.5 ± 403.6 | 16.9 ± 284.7 | 2.8 ± 1.9 | 26.9 ± 233.9 |
| **Pollen** | | | | | | | | | | |
| Timothy grass (Pheleum pratense) | 36.6 ± 57.8 | 147.0 ± 296.0 | 246.7 ± 524.8 | 157.6 ± 419.6 | 134.0 ± 350.3 | 135.7 ± 211.7 | 60.4 ± 76.0 | 17.6 ± 32.0 | 2.5 ± 1.6 | 22.6 ± 21.5 |
| Nettle (Urtica dioica) | 101.3 ± 236.7 | 227.9 ± 408.6 | 733.5 ± 595.4 | 413.4 ± 568.9 | 350.8 ± 593.1 | 287.2 ± 603.0 | 136.5 ± 581.0 | 95.2 ± 447.9 | 3.4 ± 2.3 | 29.2 ± 21.1 |
| Sheep Sorrel (Rumex acetosella) | 28.1 ± 104.6 | 258.5 ± 468.1 | 804.0 ± 605.6 | 347.4 ± 505.4 | 157.6 ± 476.1 | 137.1 ± 445.6 | 62.0 ± 213.8 | 64.7 ± 32.5 | 4.2 ± 2.1 | 32.6 ± 40.4 |
| Ryegrass (Lolium perenne) | 121.6 ± 206.6 | 307.6 ± 551.8 | 391.3 ± 611.6 | 276.4 ± 600.7 | 244.0 ± 596.6 | 209.1 ± 501.0 | 136.8 ± 454.8 | 31.7 ± 375.5 | 4.6 ± 2.7 | 38.3 ± 77.9 |
| White Poplar (Populus alba) | 75.9 ± 570.0 | 398.9 ± 464.8 | 382.2 ± 577.1 | 468.1 ± 554.9 | 330.4 ± 604.4 | 201.4 ± 581.8 | 103.5 ± 478.2 | 51.6 ± 267.3 | 3.4 ± 2.5 | 45.6 ± 1214.1 |
| European white birch (Betula pendula) | 61.6 ± 163.7 | 409.6 ± 378.6 | 712.9 ± 637.3 | 390.2 ± 523.8 | 192.3 ± 499.1 | 118.9 ± 535.3 | 66.0 ± 645.3 | 38.0 ± 525.4 | 5.1 ± 2.1 | 40.5 ± 34.2 |
| Olive (Olea europaea) | 53.8 ± 98.4 | 256.4 ± 297.4 | 774.9 ± 588.1 | 207.5 ± 473.4 | 161.3 ± 415.0 | 122.4 ± 450.8 | 66.0 ± 431.7 | 47.4 ± 461.0 | 3.1 ± 2.5 | 20.0 ± 14.5 |
| English Oak (Quercus robur) | 49.2 ± 100.3 | 182.6 ± 489.6 | 649.8 ± 562.7 | 171.1 ± 478.4 | 105.8 ± 353.3 | 45.7 ± 440.4 | 35.9 ± 485.7 | 34.3 ± 618.0 | 2.7 ± 2.9 | 26.7 ± 25.8 |
| **Fungal spores** | | | | | | | | | | |
| Cladosporium herbarium | 49.2 ± 109.8 | 238.2 ± 326.5 | 858.1 ± 579.5 | 335.4 ± 423.9 | 147.5 ± 251.2 | 92.0 ± 160.6 | 46.9 ± 91.5 | 23.0 ± 91.4 | 3.5 ± 2.1 | 23.5 ± 12.0 |
| Alternaria alternata | 29.6 ± 73.2 | 92.3 ± 249.3 | 198.3 ± 445.2 | 166.0 ± 380.3 | 141.0 ± 322.2 | 93.2 ± 224.2 | 47.9 ± 126.7 | 27.1 ± 51.0 | 2.2 ± 2.3 | 23.7 ± 30.8 |
| **Bacteria** | | | | | | | | | | |
| Esherishia coli (E. coli) | 33.1 ± 70.2 | 113.6 ± 256.3 | 328.0 ± 488.2 | 253.7 ± 441.0 | 209.6 ± 407.0 | 129.5 ± 305.0 | 56.6 ± 142.8 | 32.0 ± 52.7 | 1.4 ± 0.5 | 17.5 ± 9.7 |
| Bacillus atrophaeus (BG) (washed) | 21.7 ± 22.6 | 43.6 ± 64.3 | 120.4 ± 165.9 | 41.6 ± 63.4 | 40.9 ± 42.6 | 25.9 ± 40.4 | 15.4 ± 31.1 | 15.4 ± 18.4 | 1.1 ± 0.5 | 13.6 ± 15.6 |
| Bacillus atrophaeus (BG) (unwashed) | 16.5 ± 26.2 | 47.9 ± 93.1 | 153.5 ± 317.4 | 186.2 ± 382.9 | 176.4 ± 372.9 | 117.6 ± 282.1 | 47.1 ± 114.8 | 24.9 ± 40.1 | 1.3 ± 0.5 | 14.2 ± 371.5 |
| Bacillus atrophaeus (BT) (washed) | 18.0 ± 29.3 | 42.1 ± 72.4 | 109.4 ± 164.3 | 50.0 ± 88.4 | 38.4 ± 68.8 | 38.7 ± 53.8 | 16.6 ± 28.5 | 17.5 ± 19.7 | 1.4 ± 0.6 | 14.5 ± 8.4 |
| Bacillus atrophaeus (BT) (unwashed) | 21.0 ± 29.7 | 37.8 ± 90.7 | 109.5 ± 236.0 | 175.6 ± 363.3 | 229.4 ± 424.2 | 164.4 ± 353.4 | 69.2 ± 145.8 | 25.6 ± 38.8 | 1.3 ± 0.5 | 14.3 ± 8.9 |
| **Others** | | | | | | | | | | |
| Arizona test dust | 13.8 ± 91.9 | 31.9 ± 117.6 | 38.1 ± 274.0 | 31.5 ± 191.6 | 34.1 ± 229.5 | 19.2 ± 354.5 | 18.2 ± 226.0 | 25.2 ± 296.3 | 2.0 ± 1.8 | 19.9 ± 12.3 |
| PBS | 14.8 ± 134.4 | 19.9 ± 207.0 | 52.2 ± 395.0 | 40.5 ± 313.2 | 27.8 ± 302.5 | 27.4 ± 116.8 | 11.0 ± 35.5 | 16.3 ± 22.4 | 1.4 ± 0.8 | 16.6 ± 8.6 |
| Salt (NaCl) | 17.6 ± 20.8 | 20.9 ± 31.4 | 29.5 ± 33.0 | 25.6 ± 38.9 | 18.2 ± 27.6 | 21.4 ± 29.7 | 15.9 ± 17.9 | 16.6 ± 28.1 | 1.7 ± 1.1 | 16.0 ± 9.4 |
| Mixture | 18.4 ± 29.1 | 43.4 ± 80.4 | 113.2 ± 223.4 | 136.6 ± 281.7 | 181.6 ± 418.2 | 142.3 ± 621.8 | 67.7 ± 364.7 | 35.6 ± 96.4 | 1.3 ± 0.8 | 16.1 ± 9.2 |

**Table 4.** Median fluorescence profiles, and median particle size and shape values for the WIBS-4D.

| WIBS-4D | FL1 | FL2 | FL3 | SIZE (μm) | AF |
|---|---|---|---|---|---|
| PSLs | | | | | |
| 3 μm | 83.3 ± 189.9 | 45.4 ± 139.2 | 27.6 ± 152.3 | 1.1 ± 1.2 | 7.1 ± 5.6 |
| 2 μm blue | 58.3 ± 96.3 | 1954.4 ± 535.5 | 1806.6 ± 416.3 | 1.8 ± 0.5 | 6.2 ± 5.0 |
| 1 μm blue | 7.3 ± 13.6 | 1954.4 ± 63.9 | 1806.6 ± 31.6 | 0.9 ± 0.2 | 13.4 ± 5.7 |
| 3 μm green | 386.3 ± 86.1 | 1954.4 ± 209.3 | 1806.6 ± 48.4 | 3.0 ± 0.5 | 4.4 ± 2.8 |
| 2 μm green | 108.3 ± 31.7 | 1954.4 ± 102.3 | 1806.6 ± 37.2 | 1.9 ± 0.3 | 5.8 ± 3.7 |
| 1 μm green | 21.3 ± 10.8 | 1954.4 ± 70.5 | 1735.6 ± 138.8 | 0.9 ± 0.2 | 13.9 ± 5.7 |
| 23 μm | 359.8 ± 347.6 | 97.9 ± 148.6 | 63.6 ± 160.4 | 3.1 ± 1.8 | 9.2 ± 15.9 |
| 11 μm | 38.3 ± 474.2 | 89.4 ± 439.6 | 107.6 ± 487.2 | 1.1 ± 2.9 | 12.5 ± 14.4 |
| Pollens (complete) | | | | | |
| Timothy grass (Pheleum pratense) | 33.3 ± 288.5 | 118.4 ± 506.4 | 184.6 ± 573.6 | 1.7 ± 4.0 | 23.0 ± 19.6 |
| Nettle (Urtica dioica) | 97.3 ± 623.6 | 423.4 ± 749.3 | 620.6 ± 718.9 | 2.4 ± 5.5 | 25.1 ± 18.0 |
| Sheep Sorrel (Rumex acetosella) | 53.3 ± 516.4 | 238.4 ± 652.0 | 357.6 ± 646.2 | 2.9 ± 4.5 | 24.9 ± 18.0 |
| Ryegrass (Lolium perenne) | 77.3 ± 555.7 | 436.4 ± 727.5 | 437.6 ± 689.9 | 2.7 ± 5.2 | 24.7 ± 19.3 |
| White Poplar (Populus alba) | 34.3 ± 539.4 | 378.4 ± 683.7 | 334.6 ± 721.8 | 1.5 ± 4.8 | 20.4 ± 17.0 |
| European white birch (Betula pendula) | 83.3 ± 508.5 | 191.4 ± 552.9 | 297.6 ± 612.9 | 2.6 ± 4.1 | 26.1 ± 18.2 |
| Olive (Olea europaea) | 202.3 ± 547.9 | 87.4 ± 536.0 | 309.6 ± 585.5 | 1.1 ± 3.9 | 12.6 ± 15.6 |
| English Oak (Quercus robur) | 10.3 ± 504.4 | 91.4 ± 491.2 | 608.6 ± 650.2 | 1.1 ± 2.9 | 11.9 ± 11.5 |
| Fungal spores | | | | | |
| Cladosporium herbarium | 110.3 ± 247.7 | 210.4 ± 319.2 | 213.6 ± 331.4 | 2.3 ± 2.2 | 29.3 ± 20.5 |
| Alternaria alternata | 29.3 ± 155.1 | 84.4 ± 451.7 | 202.6 ± 594.3 | 1.2 ± 2.5 | 19.9 ± 18.9 |
| Bacteria | | | | | |
| Esherishia coli (E. coli) (unwashed) | 46.3 ± 149.7 | 228.4 ± 472.4 | 257.6 ± 529.1 | 1.0 ± 0.4 | 11.6 ± 5.7 |
| Bacillus atrophaeus (BG) (washed) | 35.3 ± 30.4 | 19.4 ± 65.6 | 29.6 ± 132.7 | 0.9 ± 0.3 | 12.7 ± 5.5 |
| Bacillus atrophaeus (BG) (unwashed) | 13.3 ± 44.8 | 163.4 ± 404.6 | 191.6 ± 462.4 | 1.0 ± 0.4 | 11.7 ± 5.7 |
| Bacillus atrophaeus (BT) (washed) | 32.3 ± 27.5 | 23.4 ± 49.4 | 30.6 ± 103.3 | 1.1 ± 0.3 | 11.5 ± 5.2 |
| Bacillus atrophaeus (BT) (unwashed) | 8.3 ± 26.7 | 300.4 ± 495.4 | 247.6 ± 485.2 | 1.0 ± 0.3 | 12.0 ± 5.8 |
| Others | | | | | |
| Arizona test dust (ATD) | 1.3 ± 241.8 | 52.4 ± 343.1 | 96.6 ± 413.1 | 1.9 ± 2.4 | 23.6 ± 16.1 |
| Phosphate-buffered saline (PBS) | 3.3 ± 123.5 | 120.4 ± 260.9 | 35.1 ± 46.0 | 1.6 ± 1.0 | 12.7 ± 6.8 |
| Salt (NaCl) | 0.3 ± 20.4 | 55.9 ± 53.0 | 39.6 ± 41.0 | 1.1 ± 0.9 | 11.0 ± 9.6 |
| Mixture | 17.3 ± 35.3 | 214.4 ± 572.6 | 209.6 ± 515.7 | 0.9 ± 0.3 | 12.7 ± 6.4 |

**Table 5.** Median fluorescence profiles, and median particle size and shape values for the WIBS-4M.

| WIBS-4M | FL1 | FL2 | FL3 | SIZE (μm) | AF |
|---|---|---|---|---|---|
| PSLs | | | | | |
| 3 μm | 312.1 ± 146.7 | 6.1 ± 22.3 | 2.9 ± 56.4 | 2.4 ± 1.8 | 4.9 ± 3.8 |
| 2 μm blue | 50.1 ± 76.6 | 2063.1 ± 490.0 | 1962.9 ± 322.6 | 1.6 ± 0.4 | 5.2 ± 4.6 |
| 1 μm blue | 4.1 ± 6.6 | 821.1 ± 168.1 | 1891.9 ± 135.9 | 0.7 ± 0.2 | 6.3 ± 2.3 |
| 3 μm green | 277.1 ± 35.7 | 2063.1 ± 99.0 | 1962.9 ± 162.2 | 2.4 ± 0.3 | 5.0 ± 1.9 |
| 2 μm green | 77.1 ± 19.4 | 2063.1 ± 48.2 | 1962.9 ± 93.2 | 1.2 ± 0.3 | 5.1 ± 3.4 |
| 1 μm green | 13.1 ± 6.9 | 420.1 ± 88.8 | 237.9 ± 51.7 | 0.8 ± 0.2 | 5.8 ± 2.3 |
| 23 μm | 256.1 ± 248.2 | 7.1 ± 14.9 | 9.9 ± 10.7 | 2.5 ± 1.1 | 6.9 ± 12.8 |
| 11 μm | 2099.1 ± 976.8 | 160.6 ± 511.5 | 70.9 ± 543.7 | 4.9 ± 5.6 | 6.9 ± 13.3 |
| Pollens (complete) | | | | | |
| Timothy grass (Pheleum pratense) | 113.1 ± 773.7 | 124.1 ± 852.7 | 198.4 ± 847.7 | 3.9 ± 7.2 | 21.3 ± 19.7 |
| Nettle (Urtica dioica) | 1658.6 ± 814.4 | 2063.1 ± 912.6 | 1962.9 ± 833.8 | 18.8 ± 8.5 | 24.6 ± 15.1 |
| Sheep Sorrel (Rumex acetosella) | 551.1 ± 923.4 | 222.1 ± 872.1 | 233.9 ± 846.0 | 5.9 ± 8.0 | 21.5 ± 17.6 |
| Ryegrass (Lolium perenne) | 713.1 ± 905.4 | 277.1 ± 894.7 | 408.9 ± 867.0 | 5.8 ± 8.3 | 23.1 ± 17.2 |
| White Poplar (Populus alba) | 422.1 ± 921.3 | 151.6 ± 843.0 | 203.9 ± 838.2 | 4.9 ± 7.6 | 20.6 ± 16.3 |
| European white birch (Betula pendula) | 209.1 ± 798.1 | 231.1 ± 856.2 | 351.9 ± 881.8 | 6.0 ± 8.7 | 25.5 ± 17.1 |
| Olive (Olea europaea) | | | | | |
| English Oak (Quercus robur) | 233.1 ± 844.4 | 86.6 ± 844.2 | 199.9 ± 749.3 | 3.0 ± 7.8 | 11.8 ± 18.2 |
| Fungal spores | | | | | |
| Cladosporium herbarium | 106.1 ± 222.7 | 36.1 ± 124.3 | 32.9 ± 162.8 | 2.6 ± 2.9 | 26.6 ± 21.1 |
| Alternaria alternata | 25.1 ± 237.2 | 30.1 ± 284.2 | 95.9 ± 416.8 | 1.3 ± 4.4 | 13.2 ± 20.5 |
| Bacteria | | | | | |
| Esherishia coli (E. coli) (unwashed) | 30.1 ± 92.5 | 34.1 ± 93.2 | 45.9 ± 150.9 | 0.9 ± 0.5 | 5.9 ± 2.5 |
| Bacillus atrophaeus (BG) (washed) | 28.1 ± 21.8 | 3.1 ± 7.4 | 6.9 ± 15.7 | 0.7 ± 0.4 | 5.9 ± 2.6 |
| Bacillus atrophaeus (BG) (unwashed) | 11.1 ± 39.0 | 28.1 ± 80.8 | 36.9 ± 113.8 | 0.9 ± 0.5 | 5.7 ± 2.4 |
| Bacillus atrophaeus (BT) (washed) | 24.1 ± 19.4 | 4.1 ± 9.1 | 5.9 ± 12.6 | 1.0 ± 0.4 | 5.8 ± 2.7 |
| Bacillus atrophaeus (BT) (unwashed) | 6.1 ± 48.8 | 47.1 ± 116.8 | 41.9 ± 127.8 | 0.9 ± 0.5 | 5.8 ± 2.4 |
| Others | | | | | |
| Arizona test dust (ATD) | 4.1 ± 491.3 | 13.1 ± 326.3 | 44.9 ± 387.7 | 2.2 ± 5.4 | 15.1 ± 16.2 |
| Phosphate-buffered saline (PBS) | 2.6 ± 202.7 | 7.1 ± 113.3 | 14.9 ± 128.3 | 1.3 ± 0.9 | 7.5 ± 8.4 |
| Salt (NaCl) | 0.1 ± 2.2 | 4.1 ± 42.7 | 16.9 ± 25.5 | 1.3 ± 1.2 | 7.8 ± 11.7 |
| Mixture | 14.1 ± 100.7 | 31.1 ± 166.4 | 43.9 ± 137.4 | 0.9 ± 1.1 | 5.8 ± 3.8 |

**Table 6.** Median fluorescence profiles, and median particle size and shape values for the WIBS-NEO.

| WIBS-NEO | FL1 | FL2 | FL3 | SIZE (µm) | AF |
|---|---|---|---|---|---|
| PSLs | | | | | |
| 3 µm | 949,644.1 ± 1,124,295.0 | 18,072.8 ± 91,258.2 | 6,957.8 ± 44,726.5 | 1.4 ± 1.5 | 4.4 ± 15.1 |
| 2 µm blue | 285,459.1 ± 959,733.5 | 14,661,725.8 ± 4,806,105.5 | 31,613,101.8 ± 9,115,405.0 | 2.4 ± 0.9 | 4.1 ± 7.3 |
| 1 µm blue | 111,156.1 ± 2,240,213.6 | 1,228,305.8 ± 366,340.7 | 2,837,835.8 ± 808,337.6 | 1.1 ± 0.2 | 5.9 ± 3.5 |
| 3 µm green | 2,802,084.1 ± 720,841.7 | 24,688,733.8 ± 3,960,189.6 | 7,907,798.8 ± 1,398,652.5 | 3.6 ± 0.5 | 4.1 ± 4.5 |
| 2 µm green | 343,505.1 ± 305,872.0 | 9,877,597.8 ± 1,677,830.1 | 3,186,115.8 ± 603,016.7 | 2.2 ± 0.5 | 4.0 ± 5.0 |
| 1 µm green | 84,830.1 ± 3,416,291.3 | 931,367.8 ± 249,472.0 | 272,009.8 ± 88,497.9 | 1.1 ± 0.2 | 4.5 ± 2.4 |
| 23 µm | 8,751,868.1 ± 8,404,649.6 | 26,167.8 ± 78,786.4 | 23,797.8 ± 61,137.1 | 3.7 ± 1.4 | 5.8 ± 10.7 |
| 11 µm | 508,279,832.1 ± 205,576,657.8 | 223,525.8 ± 1,396,336.4 | 32,137.8 ± 1,747,277.7 | 12.0 ± 3.3 | 5.8 ± 9.2 |
| Pollen | | | | | |
| Timothy grass (Pheleum pratense) | 1,624,865.1 ± 83,236,645.9 | 205,293.8 ± 32,108,331.7 | 170,059.8 ± 5,338,021.6 | 2.8 ± 5.2 | 13.8 ± 13.9 |
| Nettle (Urtica dioica) | 86,296,748.1 ± 52,817,534.7 | 69,711,965.8 ± 27,976,180.0 | 5,169,435.8 ± 4,291,601.9 | 16.5 ± 5.0 | 15.7 ± 13.1 |
| Sheep Sorrel (Rumex acetosella) | 9,652,980.1 ± 131,420,881.2 | 406,330.8 ± 35,407,618.0 | 610,315.8 ± 3,402,720.3 | 7.3 ± 6.4 | 15.4 ± 14.4 |
| Ryegrass (Lolium perenne) | 41,175,556.1 ± 144,108,873.7 | 57,473,117.8 ± 49,485,478.9 | 9,724,077.8 ± 13,311,509.2 | 15.6 ± 8.4 | 16.8 ± 16.5 |
| White Poplar (Populus alba) | 48,761,290.1 ± 149,138,701.7 | 1,426,153.8 ± 38,419,997.0 | 3,307,277.8 ± 7,368,437.3 | 11.1 ± 7.0 | 15.0 ± 17.8 |
| European white birch (Betula pendula) | 64,080,214.1 ± 95,033,743.9 | 5,709,261.8 ± 36,638,859.2 | 3,614,843.8 ± 3,228,645.1 | 14.1 ± 7.2 | 17.0 ± 8.8 |
| Olive (Olea europaea) | - | - | - | - | - |
| English Oak (Quercus robur) | 27,085,424.1 ± 219,717,163.9 | 3,872,943.8 ± 40,315,365.6 | 365,373.8 ± 4,596,854.0 | 3.6 ± 8.1 | 7.5 ± 8.6 |
| Fungal spores | | | | | |
| Cladosporium herbarium | 2,532,342.1 ± 14,374,629.0 | 96,313.8 ± 2,247,766.3 | 48,979.8 ± 1,469,403.2 | 2.7 ± 3.0 | 14.5 ± 11.9 |
| Alternaria alternata | 878,026.1 ± 12,134,583.7 | 218,701.8 ± 3,286,191.6 | 163,927.8 ± 1,704,039.2 | 1.9 ± 4.2 | 13.4 ± 11.7 |
| Bacteria | | | | | |
| Esherishia coli (E. coli) (unwashed) | 1,395,704.1 ± 6,104,494.3 | 96,149.8 ± 400,039.7 | 67,455.8 ± 319,082.8 | 1.3 ± 0.6 | 6.6 ± 7.4 |
| Bacillus atrophaeus (BG) (washed) | 577,000.1 ± 947,362.0 | 13,085.8 ± 93,188.2 | 17,012.8 ± 88,797.1 | 1.0 ± 0.4 | 6.2 ± 4.3 |
| Bacillus atrophaeus (BG) (unwashed) | 818,344.1 ± 2,189,716.3 | 77,999.8 ± 361,893.6 | 58,953.8 ± 238,205.3 | 1.4 ± 0.6 | 6.1 ± 5.1 |
| Bacillus atrophaeus (BT) (washed) | 565,390.1 ± 1,050,059.1 | 16,845.8 ± 107,775.4 | 33,609.8 ± 136,412.2 | 1.3 ± 0.6 | 7.7 ± 5.5 |
| Bacillus atrophaeus (BT) (unwashed) | 462,951.1 ± 1,598,749.9 | 82,577.8 ± 336,102.9 | 59,279.8 ± 182,927.4 | 1.2 ± 0.6 | 6.7 ± 5.0 |
| Others | | | | | |
| Arizona test dust (ATD) | 678,868.1 ± 41,490,476.3 | 150,925.8 ± 15,352,219.3 | 69,581.8 ± 1,838,486.0 | 2.3 ± 3.7 | 19.0 ± 12.8 |
| Phosphate-buffered saline (PBS) | 213,514.1 ± 1,075,310.4 | 66,635.8 ± 197,479.3 | 73,015.8 ± 287,865.8 | 0.9 ± 1.2 | 13.0 ± 15.6 |
| Salt (NaCl) | 456,328.1 ± 5,554,114.6 | 188,697.8 ± 5,669,790.9 | 195,485.8 ± 2,063,667.0 | 1.3 ± 3.1 | 9.5 ± 15.3 |
| Mixture | 709,936.1 ± 3,444,655.9 | 114,789.8 ± 1,745,612.0 | 161,639.8 ± 437,517.3 | 1.3 ± 0.9 | 5.5 ± 4.4 |

5.1.1. Particle Fragmentation

For some pollen and fungal spore samples, a bi-modal and occasional tri-modal size distribution was apparent. The data which showed the most distinct distributions were split using the lowest point between the modes as a segregation point to compare the median fluorescent profile values to the "complete" sample, as presented in Tables 2–6 in order to identify the potential effects of particle fragmentation on fluorescence characteristics.

MBS response to *Cladosporium* and *Alternaria* Material

Most instruments presented a bi-modal size distribution for the fungal spore material from *Cladosporium*, but not for *Alternaria* spores material (Figures 2 and 3). Unlike the other instruments, the MBS-D presented three distinct modes for *Cladosporium*, in addition to a slight tri-modal trend apparent for *Alternaria* material (Figures 2 and 3). Both *Cladosporium* and *Alternaria* material displayed similar signal dominance in channel XE3 when sampling particles in the first mode (<1.65 μm for *Cladosporium* and <3.75 μm for *Alternaria*). For the second and third modes, both samples showed higher fluorescence intensity in channel XE3, with second highest intensity in XE4 and third highest in XE2 (Section S8 in Supplementary Materials). When sampled from the back of the ACS, the SEM images of *Alternaria* and *Cladosporium* fungal spore material illustrate its fibrous shape and large amalgamated size (Figure 4). Given the detection limit of the instruments, smaller fragments of these larger amalgamated particles are likely to be detected. Examples of these fragments can be more clearly seen for the *Cladosporium* sample (Figure 4a).

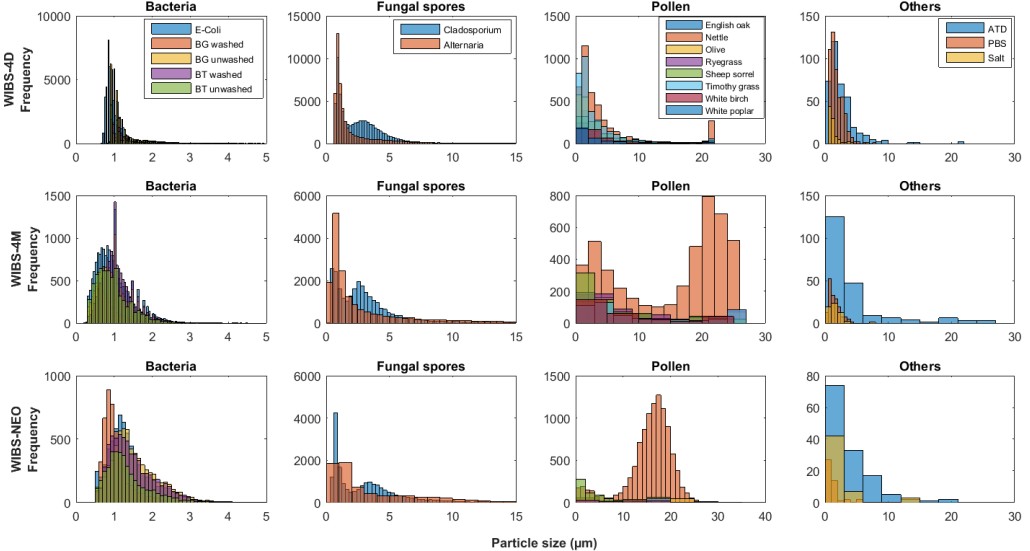

**Figure 2.** WIBS-4D, WIBS-4M and WIBS-NEO (WIBS: Wideband Integrated Bioaerosol Spectrometer) particle size histograms, split into particle type groups (bacteria, fungal spores, etc.) and presenting the size distribution of each sample.

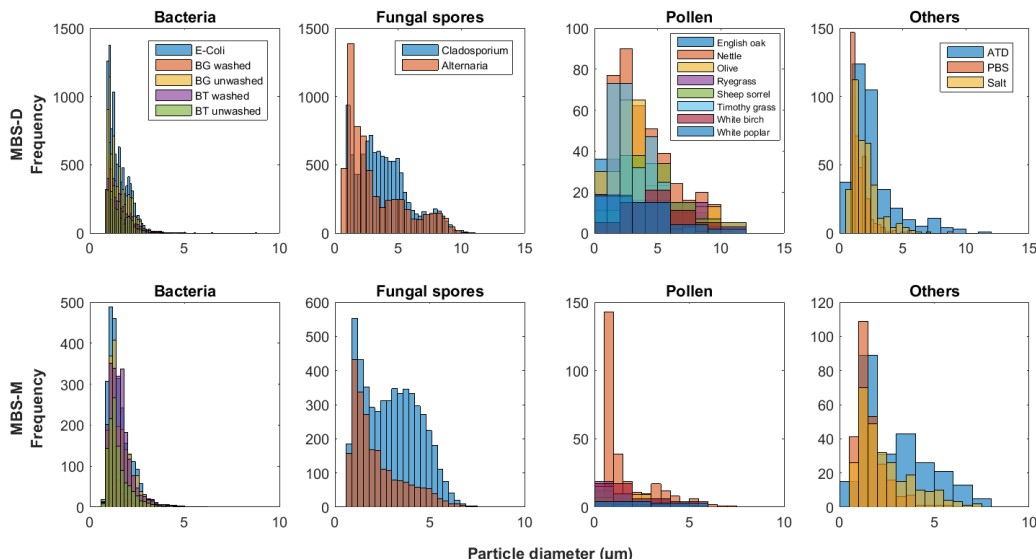

**Figure 3.** MBS-D and MBS-M (MBS: Multiparameter Bioaerosol Spectrometers) particle size histograms, split into particle type groups (bacteria, fungal spores, etc.) and presenting the size distribution of each sample.

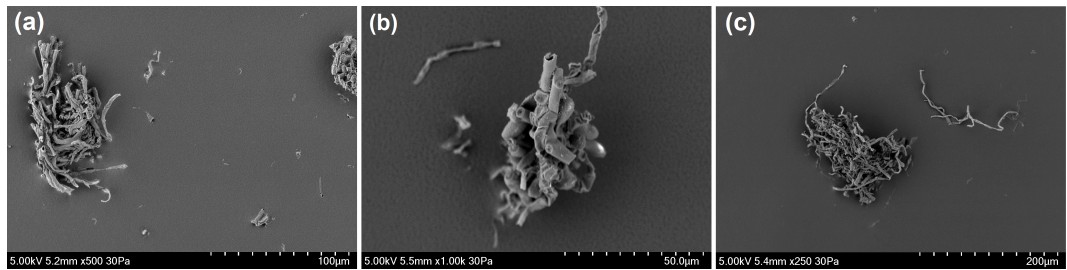

**Figure 4.** SEM (SEM: Scanning Electron Microscopy) images taken of (**a**) *Cladosporium* material and (**b**,**c**) *Alternaria* material, from the back of the ACS.

WIBS Response to Pollen Samples

The UV-LIF spectrometers with a size detection limit of ~20 μm generally do not display any clear bi-modal trends for most of the pollen sampled, with the exception of the WIBS-4M for Nettle pollen. In comparison, the WIBS-NEO, which has a larger detection size range (0.5 to ~30 μm), often presents bi-modal trends for the pollen particles sampled, with the most apparent trend present for Nettle pollen, similar to the WIBS-4M (Figure 2). When comparing the fluorescent profiles of the two modes, it can be seen that there is an increase in fluorescence signal intensity when sampling particles >13 μm (WIBS-4M) and >7.5 μm (WIBS-NEO), compared to the smaller sized fraction for the respective instruments (Table 7). Although the fluorescence profiles trend is similar for both modes sampled by the WIBS-NEO, a more apparent difference is present for the WIBS-4M, with a shift in the dominant fluorescence channel (Table 7). Although no clear fragmentation can be identified from the SEM images of Nettle pollen from both the pot and the back of the ACS, the individual particles comprise a range of shapes which would affect light scattering and detection, and influence the optical particle size of the particles sampled (Figure 5). Considering the difference in fluorescence profiles for the different size modes of the WIBS-4M for nettle pollen, this is an area potentially requiring consideration when interpreting ambient datasets as different size modes may present a varying fluorescence profile even for the same particle type detected.

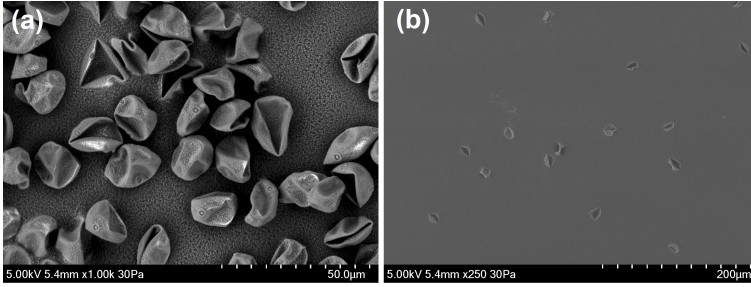

**Figure 5.** SEM images taken of Nettle pollen from: (**a**) the sample pot; and (**b**) the back of the ACS.

**Table 7.** Example of Nettle pollen bi-modality as detected by the WIBS-4M and the WIBS-NEO, presenting the median values per detection channel, particle size and shape.

| Material | FL1 | FL2 | FL3 | Size (µm) | Af |
|---|---|---|---|---|---|
| Nettle >13 µm (WIBS4M) | 1713.6 ± 790.4 | 2063.1 ± 890.3 | 1962.9 ± 815.1 | 19.4 ± 8.0 | 25.4 ± 14.7 |
| Nettle <13 µm (WIBS4M) | 178.1 ± 646.2 | 83.1 ± 510.4 | 101.9 ± 542.0 | 4.0 ± 3.2 | 24.9 ± 19.2 |
| Nettle >7.5 µm (NEO) | 89,245,846.1 ± 49,965,614.8 | 71,252,061.8 ± 23,212,541.3 | 5,267,931.8 ± 4,203,910.0 | 16.9 ± 3.2 | 15.8 ± 13.2 |
| Nettle <7.5 µm (NEO) | 2,528,544.1 ± 21,418,950.3 | 206,275.8 ± 13,599,589.4 | 182,731.8 ± 3,204,266.1 | 2.5 ± 1.9 | 14.7 ± 12.7 |

## 5.2. Particle Type Differences and Instrument Variation

To identify the detection ability of the different UV-LIF instruments and how these compare, the variations in instrument response to the broad particle type groups were initially investigated. Instrument responses to bacteria (Section 5.2.1), fungal spore material (Section 5.2.2), pollen and pollen fragments (Section 5.2.3) and non-biological samples (Section 5.2.4) were then analysed, focussing on any potential differences experienced.

Differences in fluorescence responses can be identified when comparing a selected particle type from each broad particle group (pollen, fungal spores, etc.), when sampled by the same instrument (Tables 2–6). However, the differences between the particle groups are not always consistent, and often variability is noticeable between the particle types, and depending upon the instrument used. An example of such, illustrating the difficulties in distinguishing between the different biological particle types, is presented in Figure 6, displaying a "best" and "worst" case scenario for a bacteria, fungal spore and pollen sampled by the MBS-D.

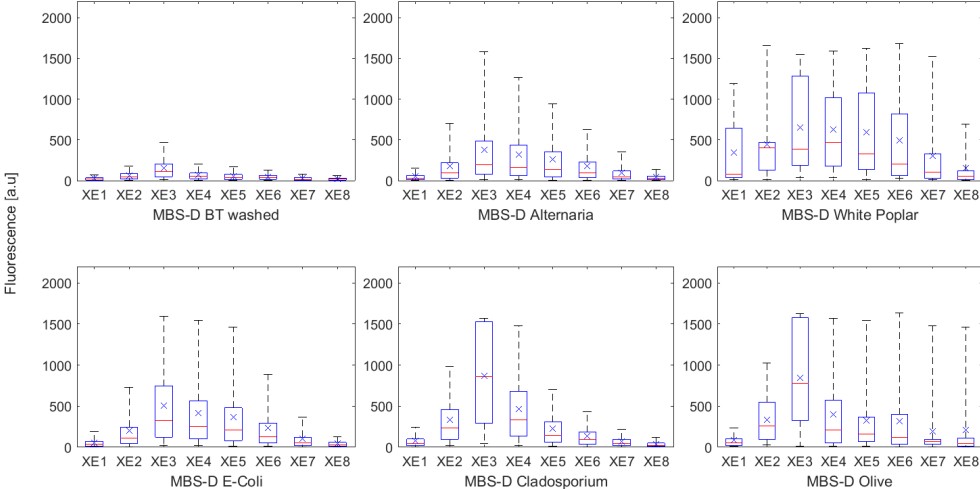

**Figure 6.** Box and whisker plots displaying the difference in fluorescence profiles for the "best" case scenario (**Top**) (BT washed, *Alternaria*, and White Poplar) and "worst' case scenario (**Bottom**) (E. coli, *Cladosporium*, and Olive), as detected by the MBS-D.

WIBS

There is a noticeable difference in fluorescence intensity, and general overall trends, when comparing the two WIBS-4 instruments to the WIBS-NEO (Tables 4–6). This results from the design of the WIBS-NEO, in which higher fluorescent intensities are detected in channel FL1, which often causes a dominance in this channel for biological particles (Table 6). Additionally, the fluorescence intensity of the WIBS-NEO ranges from ~$1 \times 10^4$ to ~$2.1 \times 10^9$, a larger range compared to the 0–2200 a.u range of the WIBS-4. Owing to the different fluorescence detection intensity of the WIBS-NEO, the fluorescence ratios of channels FL2/FL1 and FL3/FL1 for the three grouped particle types (pollens, fungal spores and bacteria) were produced to compare the general trend of the WIBS instruments (Figure 7), following Kaye et al. (2005) [10]. More particles are present for the WIBS-4D, which is likely due to the larger sample flow rate. However, the general trend between the two WIBS-4 models is in agreement, with an overlap present between fungal spores and bacteria, and a linear pollen response (Figure 7a,b). Similar to the two WIBS-4 instruments, the WIBS-NEO displays a overlap between the bacteria and fungal spore samples, but displays a three cluster divergence of pollen particles, which is suggested to result from the larger detection limit of the WIBS-NEO (Figure 7c).

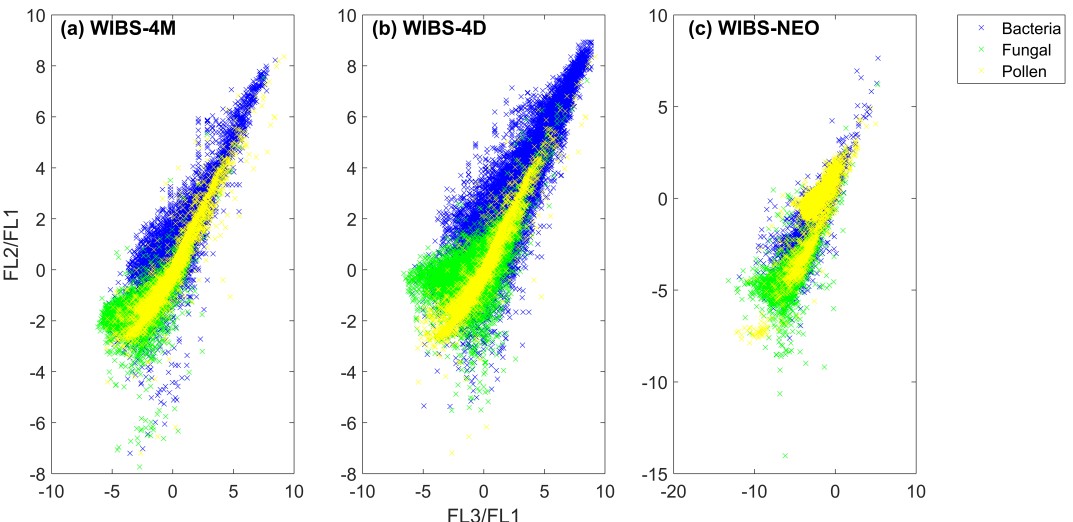

**Figure 7.** Ratio plot of fluorescence for each particle group type for: (**a**) WIBS-4M; (**b**) WIBS-4D; and (**c**) WIBS-NEO.

MBS

To compare the two MBS instruments, a ratio plot was produced for each particle group (Figure 8), following Kaye et al. (2013) [41]. Most notably, the number of particles detected by the MBS-D is greater in comparison to the MBS-M, especially for fungal spores (Figure 8). Whilst similar to the WIBS-4 instruments, few pollen particles are detected, compared to the number of bacteria and fungal spores, which likely results from the detection limit of these instruments.

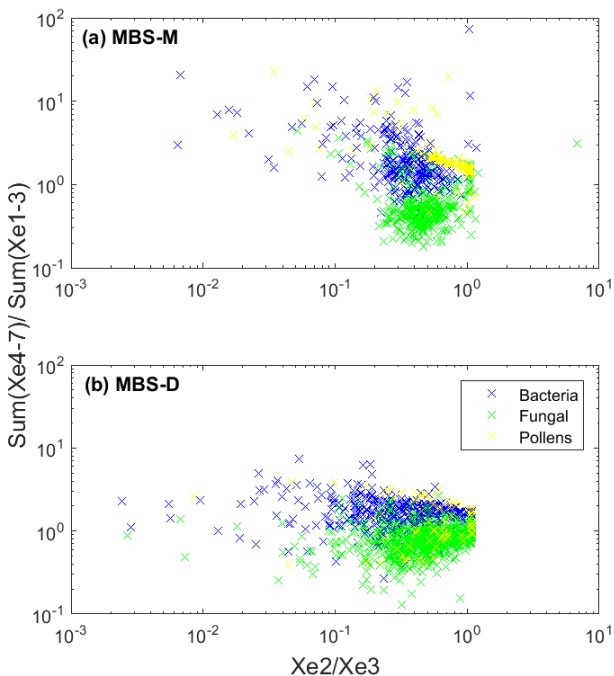

**Figure 8.** Ratio plot of channels XE2/XE3 (detection at 340–385 nm/390–435 nm) in relation to the sum of channels XE4–XE7, divided by the sum of XE1–XE3 (detection at 440–615 nm/300–435 nm) for (**a**) the MBS-M and (**b**) the MBS-D, following Kaye et al. (2013) [41], for each biological particle group type (bacteria, fungal spores and pollen).

### 5.2.1. Bacteria

The sizing of all the bacterial particle types sampled is generally consistent between the UV-LIF instruments (Figures 2 and 3), while the particle shape values produced by the WIBS-4M and WIBS-NEO are indicative of a more spherical particle shape, compared to the other instruments, especially the MBS-D (Tables 2–6).

### WIBS

Although higher fluorescence intensities can be seen for the WIBS-4D, the dominant channel for each bacterial sample is consistent when comparing the WIBS-4D and WIBS-4M (Tables 4 and 5). With the exception of unwashed BT spores, washed samples detected by the WIBS-4M and WIBS-4D display a dominance in channel FL1, compared to FL3 for unwashed samples. A similar trend was found by Savage et al. (2017) [34] and Hernandez et al. (2016) [33] who noted washed bacterial particles present a FL1 (or Type A) dominance. In comparison to the two WIBS-4 instruments, when excluding the dominant FL1 channel in the WIBS-NEO, the dominance in fluorescence ranges from FL2 for unwashed samples to FL3 for washed samples (Table 6). The number of bacterial particles remaining following the removal of the FT + 3SD baseline is higher for the WIBS-4D compared to the other WIBS models; however, this trend is notable for most samples detected and is likely due to the higher sample flow rate of the instrument (Table S3).

### MBS

The MBS-D displays higher fluorescence intensities compared to the MBS-M, though a clear consistent trend in fluorescence profiles between washed and unwashed samples cannot be identified for either instrument. Depending upon the particle type, and version of MBS (Manchester or Dstl), the dominant channel is one of XE3, XE4 and XE5. In general, the number of bacterial particle remaining

following the removal of the FT + 3SD baseline is larger for the MBS-D compared to the MBS-M, a trend apparent for all biological and non-biological samples detected (Table S3).

### 5.2.2. Fungal Spore Material

*Cladosporium* and *Alternaria* are examples of dry-discharged fungal spores, and when both fungal spore material samples are detected by the same instrument, the dominant fluorescence channel is generally consistent, with the exception of the WIBS-4M which presents a dominance in FL1 for *Cladosporium*, and FL3 for *Alternaria* (Table 5). The size distribution of the fungal material sampled is broader compared to the bacterial samples (Figures 2 and 3). In comparison to the other instruments, a clear difference can be identified in median particle shape values of *Alternaria* and *Cladosporium* material, when detected by the two WIBS-4 instruments (Tables 4 and 5).

### WIBS

A dominance in channel FL1 (Type A) was previously found by Hernandez et al. (2016) [33] for Cladosporium herbarium spores, with other samples presenting a mixture of type A, AB, and ABC fluorescence. Additionally, Savage et al. (2017) [34] found a dominance in channel FL1 for the five fungal spores sampled. On the contrary, the WIBS-4D presents higher fluorescence intensity values compared to the WIBS-4M, and shows a dominance in channel FL3 for both fungal samples (Table 4). When excluding the dominant FL1 detector for both samples detected by the WIBS-NEO, a dominance in channel FL2 is apparent (Table 6).

### MBS

Compared to the WIBS, the number of particles remaining following the removal of the FT + 3SD baseline is lower for the two MBS instruments (Table S3). Both MBS instruments show a dominance in channel XE3, with some variation present between the second and third dominant channels for the two fungal spore samples (Tables 2 and 3). The fluorescent intensity values for the MBS-D are higher than the MBS-M, and, although *Cladosporium* is more fluorescent compared to the bacterial samples (especially for the MBS-D), the same cannot be seen for *Alternaria*. Similarly, no clear differentiation can be made when comparing the fluorescence intensity of the two fungal spore samples to the bacterial samples when detected by any of the WIBS models.

### 5.2.3. Pollen and Pollen Fragments

The sizing of the pollen particles is highly variable between the instruments (Figures 2 and 3) and, compared to the manufacturer sizing by optical microscopy, most instruments present a smaller median optical scatter size value (Table S17). The difference in pollen particle sizing is likely due to the morphology of the sampled particles, as discussed for Nettle pollen (Section 5.1.1), as SEM images of Sheep Sorrel show some similar variations in particle shape and surface structure (Figure 9). Compared to the other biological particle types, the pollen samples present the most variability not only in particle size, but also in their fluorescence response.

### WIBS

For most pollen particles sampled, the WIBS-4D presents a consistent fluorescence dominance in channel FL3, with the exception of White Poplar pollen (Table 4). This trend was found by Hernandez et al. (2016) [33] for pollen particles sampled by a WIBS-4, which displayed a fluorescent dominance in channel FL3 (or Type C). In comparison, most pollen samples detected by the WIBS-4M display the highest fluorescence in channel FL1, with lower intensity in channel FL3, and lowest in channel FL2 (Table 5). A dominance in channel FL1 and also FL2 was found by Savage et al. (2017) [34] for 14 intact pollen particles sampled by the WIBS-4A, with only three out of the 14 samples dominating in channel FL3. Although there is some disparity between the fluorescent profiles of the two WIBS-4

instruments, the range of median pollen particle shape values is in agreement, and both instruments show higher fluorescence intensities compared to the bacteria and fungal spores sampled (Tables 4 and 5). The difference in fluorescent profiles between the two WIBS-4 instruments is likely due to the variance in sizing of the sampled pollen particles (Figure 2 and Tables 4 and 5).

While the WIBS-NEO has a larger detection size range, the range of median size values produced for the pollen particles sampled is similar to the WIBS-4M (Tables 5 and 6). However, the individual pollen particles sampled by the WIBS-NEO are generally larger than those sampled by the WIBS-4M, especially for Ryegrass pollen (Tables 5 and 6). Although the WIBS-NEO presents a lower median particle shape range compared to the two WIBS-4 instruments, all of the WIBS instruments present the smallest Af values for English Oak and the largest for White Birch (Tables 5–6). Additionally, a difference in fluorescence intensity can be identified when comparing the pollen sampled by the WIBS-NEO to the bacteria and fungal spore samples. Unlike the other pollen particle types sampled by the WIBS-NEO, Ryegrass pollen is the only pollen type to fluoresce highest in channel FL2, as opposed to FL1. When excluding the dominant FL1 channel, the fluorescence trend of the pollen sampled by the WIBS-NEO is similar to the WIBS-4M for Sheep Sorrel and White Poplar pollen (Tables 5 and 6).

MBS

In comparison to the other instruments, the pollen particles sampled by the MBS-M are considerably smaller in measured size (Figure 3), with some pollen samples similar in size or smaller than some bacterial samples (Table 2). The range of median pollen sizes detected by the MBS-D is larger than that of the MBS-M and WIBS-4D (Tables 2–6). The number of particles remaining following the removal of the FT + 3SD baseline for both MBS instruments is considerably lower compared to the three WIBS instruments, with the MBS-M consistently presenting the lowest number of pollen particles remaining following the baseline removal (Table S3).

The MBS-D presents a fairly consistent fluorescence response for the pollen sampled, with most particle types dominating in channel XE3 with the exception of White Poplar (Table 3). In general, the MBS-D displays higher fluorescent intensity values for the pollen sampled compared to the bacterial samples, but shows a similar fluorescence intensity to *Cladosporium*, which may be due to the similarity in median particle size (Table 3). Compared to the MBS-D, the MBS-M presents a considerable difference in fluorescence intensities between the pollen samples and the other biological groups (Table 2). However, the fluorescence dominance trend for the pollen sampled by the MBS-M are more variable, and of the eight samples, four present a dominance in channel XE7.

Interestingly, the pollen samples detected by the MBS-M which have a median size of 0.9 μm (Nettle, Sheep Sorrel and Ryegrass) display the highest fluorescence values, with a dominance in channel XE7 (Table 2). Particles larger than 0.9 μm generally show lower fluorescent intensity values, especially noticeable when the median particle size is >2 μm (White Poplar and English Oak), which show a fluorescence dominance in channel XE1. Although there does appear to be some similarity in particle shape for some of the individual pollen particle types sampled by the MBS-M and MBS-D, the particle shape range is higher for the MBS-D compared to the MBS-M (Tables 2 and 3).

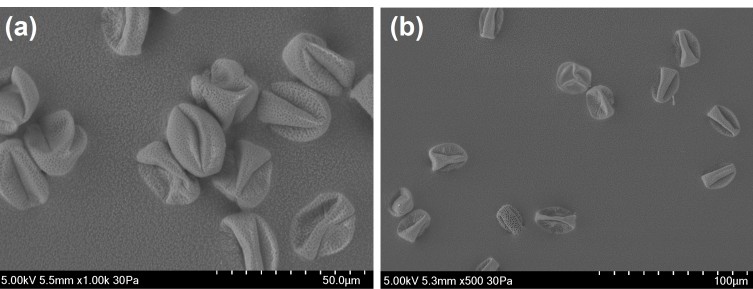

**Figure 9.** SEM images taken of Sheep's Sorrel pollen from: (**a**) the sample pot; and (**b**) the back of the ACS.

### 5.2.4. Non-Biological Samples

In addition to the biological particles, non-biological potential interferent particles were also sampled comprising Arizona Test Dust (ATD), Phosphate-buffered saline (PBS) and Salt (NaCl).

WIBS

The non-biological samples detected by the WIBS-4D present a dominance in channel FL2 for the PBS solution and salt sample, and FL3 for ATD (Table 4). Although the fluorescence intensities are not unlike the washed bacterial samples, the non-biological samples all present weak fluorescence in channel FL1, even though the median size range is somewhat similar to the bacterial samples (Table 4). The median size range of the non-biological particles sampled by the WIBS-4M are generally in agreement with the WIBS-4D, although some difference can be seen in particle shape range when comparing the WIBS-4M to the WIBS-4D (Tables 4 and 5). The non-biological particles as detected by the WIBS-4M are generally weakly fluorescent compared to the biological particles, and show a consistent dominance in channel FL3, with low intensity in channel FL1, similar to the WIBS-4D. When excluding the FL1 channel from the WIBS-NEO, the PBS solution and salt sample present a dominance in channel FL3, compared to a dominant channel FL2 for ATD (Table 6). Across the three channels, the fluorescence intensities of the WIBS-NEO for the non-biological particles are higher than that of the washed bacterial samples, an opposite trend to the two WIBS-4 instruments, even though the median particle size range is similar (Tables 4–6).

MBS

Similar to the two WIBS-4 instruments, the two MBS instruments present lower fluorescence intensity values for the non-biological samples compared to the biological samples. The sizing of the non-biological samples are reasonably consistent between the MBS-M and the MBS-D (Figure 3). With the exception of the salt sample detected by the MBS-M, the other two samples are most fluorescent in channel XE3 (Table 2). This trend is replicated by the MBS-D which presents a dominant channel XE3 for the three non-biological samples, although more variation can be seen in the other detector channels in comparison to the MBS-M (Table 3). The number of non-biological particles remaining following the removal of the FT + 3SD baseline are higher for the two MBS instruments, especially when compared to the WIBS-4M and WIBS-NEO (Table S3).

### 5.3. Relationship between Particle Size, Shape, and Fluorescence

To identify any relationships between particle size and shape, in relation to particle fluorescence, adapted ratio plots were produced for the broad particle types, as in Section 5.2. Although no clear trends could be identified for the particle size and shape ratio plots for the MBS, the larger detection ability of the WIBS-NEO is especially apparent for the pollen samples in relation to particle size, with three distinct pollen groups comprising different sizes (Figure 10a).

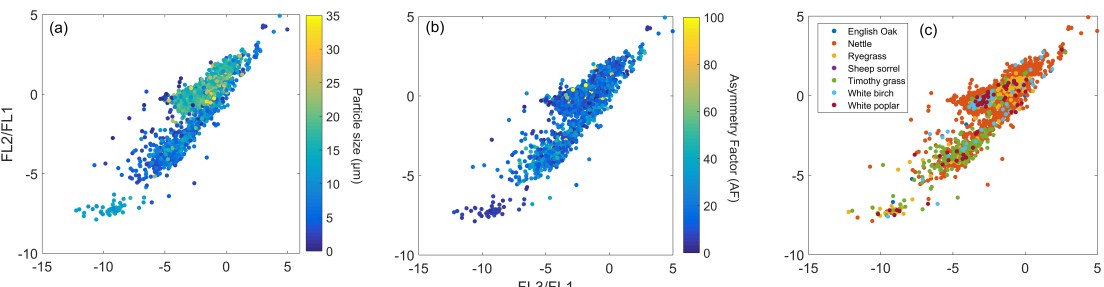

**Figure 10.** Fluorescence ratio of pollen sampled by the WIBS-NEO in relation to: (**a**) particle size; (**b**) particle shape, and (**c**) particle type.

Of the three pollen groups detected by the WIBS-NEO, one group consists of spherical pollen particles, with the lowest FL3/FL1, FL2/FL1 ratio (Figure 10b), and comprises a mixture of different pollen particle types sized around 15 μm (Figure 10a). The WIBS-4M also detects this group of mixed pollen, and presents similar size and Af values, and low FL3/FL1 and FL2/FL1 ratio (Figure S9). This group of pollen particles sampled by the WIBS-NEO shows some similarity to the adapted ratio plot of the two fungal spore materials, *Alternaria* and *Cladosporium*, in which a mixture of the two particle types comprise the smallest sized particles with the lowest Af values, but, unlike the pollen sampled by the WIBS-NEO, present a high FL3/FL1 and FL2/FL1 ratio (Figure S10). In comparison to the other instruments, the most apparent difference detected by the WIBS-NEO is the cluster of pollen particles sized around 20 μm, which comprises a mixture of spherical and aspherical particles, which are heavily dominated by Nettle pollen (Figure 10).

### 5.3.1. Relationship between Particle Size and Shape

The ability of the different UV-LIF instruments to produce similar particle size and shape values were investigated for each instrument and particle type group (Figure 11).

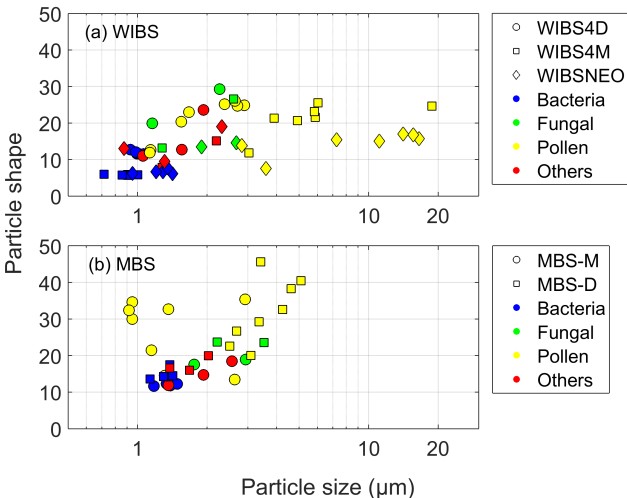

**Figure 11.** The relationship between median particle size and median particle shape for: (**a**) WIBS; and (**b**) MBS. Different markers represent the different instruments, and are coloured according to particle type group.

### WIBS

The WIBS-NEO presents a slightly larger median particle size for the bacterial samples compared to the two WIBS-4 instruments, although similar Af values for these particles can be seen when comparing the WIBS-NEO to the WIBS-4M (Figure 11a). Particles sized > 3 μm primarily consist of pollen particles sampled by the WIBS-4M and the WIBS-NEO. Although the pollen samples detected by the WIBS-4D are < 3 μm, the Af is comparable to the WIBS-4M, while the WIBS-NEO presents lower Af values for both pollen and fungal spores samples.

### MBS

The MBS presents higher median shape values compared to the WIBS for all the particles sampled. Although the pollen particles are smaller in size compared to the three WIBS instruments, there is a notable difference in pollen particle shape (Figure 11b). Unlike the other instruments, the MBS-D presents a relationship between pollen particle size and shape, in which an increase in median size results in an increase in median particle shape values. Excluding the pollen particles, both MBS instruments show relatively good consistency, with similar size and shape values between the two

instruments. In addition, some crossover similarities in particle size and shape can be seen when comparing both MBS instruments to the WIBS-4D.

Although the usefulness of using particle shape to distinguish between different particle types is uncertain, if particle size is not considered, it would be possible to distinguish most of the pollen particles sampled by the MBS, as most pollen particles sampled have a particle shape >25 (Figure 11b). Although there are a few pollen samples which have a similar median shape value to the fungal spore samples, the greater distinction between the pollen samples, and general variance in particle shape values compared to the WIBS may offer some level of particle type discrimination. This may result from the difference in particle shape detection by the MBS compared to the various WIBS models, as outlined in Sections 3 and 4.1.

*5.4. Response to Dust–Bacterial mixtures*

Numerous studies have found links between dust storms and increases in bacterial and fungal spore concentrations [4,37,38]. Using the ACS, a mixture of particles were generated comprising a 1 μm PSL, a bacteria sample and ATD. By producing a mixture of particles, the response of the different UV-LIF spectrometers could be investigated. In addition, these data can be analysed using various discrimination algorithms, to assess how such an event is interpreted. The mixture datafile was additionally split to analyse the relationship between bacteria and the ATD sample, and compare to the complete mixture file. The data were split using the recorded times of the different particle samples generated within the ACS.

WIBS

The dominant channel and median particle size values are consistent when comparing the "complete" and "split" file sampled by the three WIBS instruments (Table 8). Some variance can be seen in the second dominant fluorescence detector channel when comparing the complete and split WIBS-NEO mixture files. Compared to the individual ATD sample, only the WIBS-4D shows a difference in the dominant fluorescent channel for the complete and split mixture file (Table 8).

**Table 8.** Median particle fluorescence, size and shape for the "complete" (1μm PSL, bacteria, and ATD) (ATD: Arizona Test Dust) and "split" (Bacteria and ATD) mixture files for the three WIBS instruments (WIBS-4D, WIBS-4M, and WIBS-NEO), in addition to ATD fluorescence data.

| Sample | Instrument | FL1 | FL2 | FL3 | Size (μm) | Shape |
|--------|-----------|-----|-----|-----|-----------|-------|
| Complete | WIBS-4D | 17.3 ± 35.3 | 214.4 ± 572.6 | 209.6 ± 515.7 | 0.9 ± 0.3 | 12.7 ± 6.4 |
| Split | WIBS-4D | 21.3 ± 36.5 | 188.4 ± 356.4 | 174.6 ± 382.1 | 0.9 ± 0.3 | 12.5 ± 6.1 |
| ATD | WIBS-4D | 1.3 ± 241.8 | 52.4 ± 343.1 | 96.6 ± 413.1 | 1.9 ± 2.4 | 23.6 ± 16.1 |
| Complete | WIBS-4M | 14.1 ± 100.7 | 31.1 ± 166.4 | 43.9 ± 137.4 | 0.9 ± 1.1 | 5.8 ± 3.8 |
| Split | WIBS-4M | 17.1 ± 50.5 | 26.1 ± 62.9 | 30.9 ± 88.2 | 0.9 ± 0.9 | 5.8 ± 3.2 |
| ATD | WIBS-4M | 4.1 ± 491.3 | 13.1 ± 326.3 | 44.9 ± 387.7 | 2.2 ± 5.4 | 15.1 ± 16.2 |
| Complete | WIBS-NEO | 709,936.1 ± 3,444,655.9 | 114,789.8 ± 1,745,612.0 | 161,639.8 ± 437,517.3 | 1.3 ± 0.9 | 5.5 ± 4.4 |
| Split | WIBS-NEO | 708,300.1 ± 1,302,565.7 | 65,899.8 ± 232,694.7 | 51,677.8 ± 371,714.5 | 1.3 ± 0.7 | 5.7 ± 4.0 |
| ATD | WIBS-NEO | 678,868.1 ± 41,490,476.3 | 150,925.8 ± 15,352,219.3 | 69,581.8 ± 1,838,486.0 | 2.3 ± 3.7 | 19.0 ± 12.8 |

MBS

The two MBS instruments present the same, or near similar median particle size value when comparing the complete and split file (Table 9). Although the MBS-D does not show a change in dominant channel (XE5), the dominant channel for the MBS-M does change from channel XE5 for the complete file, to XE3 for the split file (Table 9). The median size for both the complete and split mixture samples detected by the MBS instruments is smaller than the individual ATD sample, additionally the ATD sample presents a dominance in channel XE3, similar to the split MBS-M mixture (Table 9).

**Table 9.** Median particle fluorescence, size and shape for the complete (1 µm PSL, bacteria, and ATD) and split (Bacteria and ATD) mixture files for the two MBS instruments (MBS-M and MBS-D), in addition to ATD fluorescence data.

|  | Instrument | XE1 | XE2 | XE3 | XE4 | XE5 | XE6 | XE7 | XE8 | Size (µm) | Shape |
|---|---|---|---|---|---|---|---|---|---|---|---|
| Complete | MBS-M | 6.0 ± 112.8 | 18.0 ± 63.1 | 44.0 ± 121.3 | 47.2 ± 135.1 | 61.1 ± 162.9 | 46.4 ± 372.5 | 33.7 ± 163.2 | 19.6 ± 133.0 | 1.3 ± 1.0 | 12.2 ± 7.8 |
| Split | MBS-M | 5.0 ± 116.5 | 20.1 ± 90.6 | 53.7 ± 135.3 | 49.7 ± 136.0 | 48.4 ± 142.4 | 33.1 ± 112.3 | 15.1 ± 115.7 | 10.2 ± 276.4 | 1.4 ± 0.8 | 12.0 ± 6.5 |
| ATD | MBS-M | 4.2 ± 6.0 | 8.5 ± 31.0 | 12.7 ± 141.7 | 11.5 ± 160.3 | 10.9 ± 75.3 | 9.1 ± 26.1 | 8.8 ± 29.0 | 5.8 ± 9.5 | 2.6 ± 1.8 | 18.5 ± 12.2 |
| Complete | MBS-D | 18.4 ± 29.1 | 43.4 ± 80.4 | 113.2 ± 223.4 | 136.6 ± 281.7 | 181.6 ± 418.2 | 142.3 ± 621.8 | 67.7 ± 364.7 | 35.6 ± 96.4 | 1.3 ± 0.8 | 16.1 ± 9.2 |
| Split | MBS-D | 15.7 ± 37.7 | 51.4 ± 116.1 | 116.8 ± 222.8 | 142.2 ± 288.1 | 153.5 ± 299.1 | 106.3 ± 209.4 | 38.8 ± 113.8 | 18.5 ± 39.8 | 1.3 ± 0.6 | 17.2 ± 9.9 |
| ATD | MBS-D | 13.8 ± 91.9 | 31.9 ± 117.6 | 38.1 ± 274.0 | 31.5 ± 191.6 | 34.1 ± 229.5 | 19.2 ± 354.5 | 18.2 ± 226.0 | 25.2 ± 296.3 | 2.0 ± 1.8 | 19.9 ± 12.3 |

## 6. Conclusions and Recommendations

Understanding the response of different UV-LIF spectrometers to known particle types, and identifying differences between multiple versions of the same instrument is imperative for future UV-LIF applications and database comparisons. In this paper, an intercomparison of multiple UV-LIF spectrometers is presented, using a UK-based chamber facility for bioparticle measurements. By using the ACS, differences between the UV-LIF instruments, including different versions of the same instrument, were identified. The following key results are highlighted:

- The number of particles remaining following the removal of FT + 3SD presented a notable difference between the WIBS and MBS. A generally consistent greater number of interferent/non-biological samples were detected by the two MBS instruments especially when compared to the WIBS-4M and WIBS-NEO, and a considerably lower number of pollen particles were detected by the MBS instruments compared to the WIBS (Table S3).
- Pollen samples were the most variable in size and fluorescence response, and often sized smaller than expected, as shown by the WIBS-4D and both MBS instruments (Section 5.2.3). The variability in detected size is suggested to result from the influence of particle morphology affecting light scattering, as identified from SEM images of the samples. Such variability illustrates the potential difficulties in using laboratory data for ambient data interpretation.
- A clear defining trend in fluorescence response could not be identified between the different biological groups (Section 5.2). Additionally, the differences in fluorescent intensities were not always apparent, and often dependent on the instrument used. Although most non-biological particles generally presented lower fluorescence intensities compared to the biological samples, the WIBS-NEO presented higher fluorescent intensities for non-biological particles than some biological particles sampled (Section 5.2.4 and Table 6) due to the detector configuration which may require careful determination of thresholds for subsequent analysis.
- Compared to the WIBS, the MBS presented higher shape values for all particles sampled, especially for pollen particles (Section 5.3.1). To some degree, the larger variation in shape values between the particle groups would enable the pollen samples to be segregated from the other samples, making this potentially useful as an additional classification parameter.
- While only differences in fluorescence intensities could be seen for the different size modes of fragmented particles (Section 5.1.1), the WIBS-4M was the only instrument to display a different fluorescence profile for each size mode of Nettle pollen. The difference in fluorescence between the modes detected by the WIBS-4M requires consideration when interpreting ambient datasets.

The ability to produce common datasets using the ACS provides a useful method for instrument and database comparisons. However, the use of these data for further applications, such as for interpretation of ambient datasets and for discrimination algorithm development, is not clear cut. For example, dried pollen samples are unlikely to be fully representative of real-world pollen samples. In addition, the size and shape of the fungal spore material (Figure 4) are not representative of real-world fungal spores. Therefore, some caution needs to be taken when using these data in addition to other laboratory data for comparison to ambient datasets and for use as training data for algorithm development. It must also be noted that due to the differences in the detection of different material for multiple versions of nominally the same instrument (e.g., WIBS-4M and WIBS-4D), that training data are not cross-compatible, and must only be used when collected by the same model instrument.

The development of new instruments with more fluorescence detection channels, such as the PLAIR Rapid-E and the SIBS, open up the potential to further discriminate between biological particle types. Considering that some segregation could be seen between pollen particles and the other biological samples when detected by the MBS, instruments which offer the combination of higher spectral resolution and shape analysis may prove useful for future discrimination approaches.

Future work utilising these new instruments, in addition to those used in this study, within laboratory conditions, is required to build upon and compare to the results presented here. Whilst,

additionally, the standardisation of sampled particles including their generation would allow for more widely applicable instrument intercomparisons and data integration.

This work utilised a WIBS-NEO, courtesy of Droplet Measurement Technologies, the results of which are representative of the instrument version used at this point in time, and may not necessarily be reproduced when using different versions of the instrument. For future studies, and for general comparison purposes, it would serve well to document the specific version and model number of the instruments used, when available, to assess any variation between different versions of the same instrument as development continues (Table A1).

**Supplementary Materials:** The following are available online at http://www.mdpi.com/2073-4433/10/12/797/s1; Table S1: List of particle types generated and provenance; Figure S1: Schematic pre-processing data protocol flowchart; Table S2: total number of particles detected following data collection (original data length); Table S3: Number of particles remaining following FT + 3SD baseline removal; Table S4: Percentage change between total number of particles detected and number of particles remaining following FT + 3SD baseline removal; Table S5: Number of particles remaining following FT + 9SD baseline removal for MBS; Table S6: Percentage change between original data length and data with FT + 9SD removed for the MBS; Table S7: Fluorescence profiles of particles sampled on multiple days for the MBS-D; Table S8: Fluorescence profiles of particles sampled on multiple days for the MBS-M; Table S9: Fluorescence profiles of particles sampled on multiple days for the WIBS-4M; Table S10: Fluorescence profiles of particles sampled on multiple days for the WIBS-4D; Table S11: Fluorescence profiles of particles sampled on multiple days for the WIBS-NEO; Table S12: Comparison between the fluorescence profiles of the 2019 data following the repair of the WIBS-4D to the 2017 ACS data; Table S13: Comparison between the 2017 WIBS-4D and 2019 WIBS-4D data; Figure S2: MBS-M Average FT plot per file; Figure S3: MBS-D Average FT plot per file; Figure S4: *Cladosporium* and *Alternaria* tri-modal pattern for MBS-D; Table S14: Fungal spore MBS-D tri-modality values; Figure S5: *Cladosporium* tri-modal pattern for the MBS-D plot; Figure S6: *Alternaria* tri-modal pattern for the MBS-D plot; Table S15: WIBS bi-modal profiles for *Cladosporium*; Table S16: MBS-M bi-modal profiles for *Cladosporium*; Figure S7: *Cladosporium* bi-modal pattern for the WIBS-4M and WIBS-4D; Figure S8: *Cladosporium* bi-modal pattern for the WIBS-NEO and MBS-M; Table S17: UV-LIF instrument median size values compared to Greer sizing; Figure S9: Ratio plot of pollen sampled by the WIBS-4M of particle size, particle shape and particle type; Figure S10: Ratio plot of fungal spores sampled by the WIBS-NEO of particle size, particle shape, and particle type.

**Author Contributions:** Data curation, E.F.; Formal analysis, E.F., M.G. and D.T.; Funding acquisition, M.G., V.F., R.S.-E. and D.T.; Investigation, E.F., M.W., V.F., A.A. and G.G.; Resources, M.W. and V.F.; Software, E.F. and D.T.; Supervision, M.G. and D.T.; Visualisation, E.F., M.G. and D.T.; Writing—original draft, E.F., M.G. and D.T.; and Writing—review and editing, E.F., M.G., M.W., V.F., A.A., R.S.-E., W.S., P.K. and D.T.

**Funding:** E.F. is funded under the Dstl (Defence Science and Technology Laboratory) and DGA (Direction Générale de l'Armement) Anglo-French PhD scheme (Grant reference DSTLX-1000120837) and affiliated to the NERC EAO Doctoral Training Partnership.

**Acknowledgments:** The authors would like to thank Sarah Cordery (Dstl) and Simon Smith (Dstl) for the SEM analysis conducted.

**Conflicts of Interest:** The authors declare no conflict of interest.

## Appendix A. Instrument Version

**Table A1.** Software and firmware versions of the UV-LIF instruments used in this study.

| Instrument | Software Version | Firmware Version |
| --- | --- | --- |
| WIBS-4M | N/a | N/a |
| WIBS-4D | N/a | N/a |
| WIBS-NEO | 2.3.3.16 | 42 |
| MBS-M | 4.3.0.3 | N/a |
| MBS-D | 4.5.0.9 | N/a |

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
