# Peer review of "Intercomparison of Multiple UV-LIF Spectrometers Using the Aerosol Challenge Simulator"

_atmosphere, doi:10.3390/atmos10120797_

Round 1

Reviewer 1 Report

In this work, the authors used an Aerosol Challenge Simulator (ACS) with controlled concentrations of biological (e.g. bacteria, fungal spores and pollen)and non-biological particles (e.g. dust, saline and salt) to test and intercompare the performance of various UV-LIF spectrometers. The paper is very well written and provides new insight and thoroghful discussion of how various factors determine the instrument performance and variations. The recommendations suggested by the authors are highly valuable for the investigators and general readers. This is a very high quality paper. I am strongly recommended the publication of this article in Atmosphere in present form.

Author Response

We would like to thank reviewer one for their positive comments and review of the paper.

Reviewer 2 Report

As someone who has extensive experience of sampling and enumerating bioaerosols and who is aware of the variation between different sampling methods, I was very interested in this paper given the title and the introduction. the introduction provides a good overview of the methods being compared and identifies the lack of robust laboratory studies comparing the performance of these real-time options.

The methodology appears to be appropriate and is described well in the methodology section and consists of a controlled bioaerosol generation test and simultaneous detection with the five instruments.

The results are extensive and form the largest section of the paper, however a number of the tables and figures are too small for the reader to easily interpret and I personally found the results section difficult to follow and remain focussed.

As someone who uses samplers extensively, what I was expecting after the results section was some kind of narrative to explain how the results show what the comparative performance of the instruments was. How do the different instruments compare in terms of their ability to detect and enumerate the different types of bioaerosols. Are the concentrations comparable to each other or is there significant variations and if so why? How well do each of the instruments distinguish between biological and non-biological particles?

This narrative was absent and it was difficult to see how the conclusions had been reached. Overall I came away having read the paper with a sense of disappointment as I didn't feel it had delivered what I expected it to deliver.

Author Response

We would like to thank reviewer two for their careful review and their constructive comments which we have now incorporated and which will improve the quality of the manuscript. We are sorry to hear that reviewer two felt that the paper did not appear to deliver what was expected from their perspective and we hope that the changes made, specifically in regards to providing a stronger narrative throughout the results section, improves this.

Firstly, with regards to the comment on the small size of some of the figures and tables within the paper, where possible we have made changes to those that we thought may be more difficult to read, and hope that this is sufficient at this stage. For larger tables, such as those containing data from the MBS, we will consult with the journal editor for the protocol in presenting such wide tables.

In terms of the main comments on lack of narrative or focus within the results section, we have produced a clearer narrative as we acknowledge that the results section is very extensive, and a clearer structure would allow for easier interpretation of the many results produced by this work.  Though we have not added a separate section following the results section, we have re-worked the narrative within the results section, including a pre-discussion section, to illustrate how the final conclusions had been reached.

An overview of the changes made within the results section is presented below:

The structure of the results section has been altered to reflect the four individual parts and their results produced from the data analysis, with four clear subsections now presented. An overview of the structure of the results section is now introduced prior to the presentation and discussion of the results highlighting the four main subsections that make up the main results section in order to signpost the reader to the different parts of analysis conducted. We have focused on linking our main conclusions to the relevant sections in which the main conclusions are taken from, so that these concluding remarks can be clearly linked back to their presentation and discussion within the appropriate results sections. In response to the points ‘How do the different instruments compare in terms of their ability to detect and enumerate the different types of bioaerosols’ and ‘How well do each of the instruments distinguish between biological and non-biological particles?’ The description in Section 5.2. Particle type differences and instrument variation, presents a discussion of the noted differences between the different particle groups and instrument responses in each respective section. Additional descriptions have now been added to build upon this. In response to the question ‘Are the concentrations comparable to each other or is there significant variations and if so why?’ Clarification within the conclusion section and section 5.2 has been added to state the general and notable trends in the number of particles remaining following the removal of the FT + 3SD baseline as presented in supplementary Table S3.

Reviewer 3 Report

The manuscript by Forde et al. provides a technical summary of using an ACS to compare different WIBS/MBS insturments. I think this manuscript is appropriate for publication in Atmosphere and I have only a few minor comments:

1) I think it is important to acknowledge/discuss in the introduction that none of these instruments are picking up viruses, which are very important bioaerosols.

2) Can the authors please describe what controls (specifically negative) were used?

3) In a real-world scenario most bioaerosols are not single particles, rather they are clustered or attached to carrier particles. Does this have any affect on the instrument performance?

4) Although there is a lot of interesting data presented, I feel as though the manuscript is too longer (and difficult to follow in some sections). It would be nice to see a more concise paper. For example, section 3.1 seems unnecessarily long and I feel as though much of the information can be put into a table, which would be easier to understand.

Author Response

We would like to thanks reviewer 3 for their review of the manuscript who suggested minor comments for improvement of this work. In response to the suggestions, a description of the changes made to the manuscript is outlined below:

‘I think it is important to acknowledge/discuss in the introduction that none of these instruments are picking up viruses, which are very important bioaerosols.’ This has been added within the introduction section to highlight what particle types are within the detection range of these instruments and that these instruments are able to detect bacteria, fungal spores, and pollen/pollen fragments, but not smaller particles such as viruses (Page 2; lines 46-47). ‘Can the authors please describe what controls (specifically negative) were used?’ A range of non-biological but potentially fluorescent particles were used alongside the biological samples in addition to calibration PSLs, similar to previous laboratory studies conducted (e.g. Savage et al 2017; Hernandez et al 2016). ‘In a real-world scenario most bioaerosols are not single particles, rather they are clustered or attached to carrier particles. Does this have any affect on the instrument performance?’ To try and replicate this, a mixture of bacteria, Arizona Test Dust (ATD), and 1um diameter polystyrene latex spheres (PSL) were generated using the ACS, as presented in Section 5.4. As previous studies have highlighted the links between dust and airborne bacteria, the influence of the ATD and bacteria was investigated separately from the bacteria/ATD/PSL mixture produced by using the start and stop times recorded during the experiment to select the period in which only dust and bacteria were generated within the chamber. The fluorescent profiles were then compared against the dust sample to identify the influence of the bacteria (listed as ‘split’ within Table 3 and Table 4) and bacteria and 1um PSL’s together (listed as ‘complete’ within Table 3 and Table 4). For the WIBS-4D, and MBS-M there is an apparent difference in fluorescence intensity, with higher intensity for the complete and split sample data, compared to the sampled ATD. However, this is a trend which is not consistent when we compared the fluorescence between the detector channels for the other instruments. Following this test mixture, produced using the ACS, it is suggested that further work is conducted focusing on producing and interpreting such mixtures to further identify how these instruments respond to such events which may be more environmentally relevant. ‘Although there is a lot of interesting data presented, I feel as though the manuscript is too longer (and difficult to follow in some sections). It would be nice to see a more concise paper. For example, section 3.1 seems unnecessarily long and I feel as though much of the information can be put into a table, which would be easier to understand.’ We acknowledge that this is a lengthy paper and have considerably shortened the suggested section 3.1. Following comments received from reviewer 2, we have also restructured the results section and made sure to provide an results section overview prior to delving into the results in greater detail in order to signpost the reader to sections which may be of particular interest to them.

Round 2

Reviewer 2 Report

Thank you for addressing my initial concerns regarding the presentation and explanation of your results.